# Expansion of the circadian transcriptome in *Brassica rapa* and genome-wide diversification of paralog expression patterns

Kathleen Greenham[1][†][*], Ryan C Sartor[2][†], Stevan Zorich[1], Ping Lou[3], Todd C Mockler[4], C Robertson McClung[3][*]

[1]Department of Plant and Microbial Biology, University of Minnesota, Saint Paul, United States; [2]Crop and Soil Sciences, North Carolina State University, Raleigh, United States; [3]Department of Biological Sciences, Dartmouth College, Hanover, United States; [4]Donald Danforth Plant Science Center, St. Louis, United States

**Abstract** An important challenge of crop improvement strategies is assigning function to paralogs in polyploid crops. Here we describe the circadian transcriptome in the polyploid crop *Brassica rapa*. Strikingly, almost three-quarters of the expressed genes exhibited circadian rhythmicity. Genetic redundancy resulting from whole genome duplication is thought to facilitate evolutionary change through sub- and neo-functionalization among paralogous gene pairs. We observed genome-wide expansion of the circadian expression phase among retained paralogous pairs. Using gene regulatory network models, we compared transcription factor targets between *B. rapa* and Arabidopsis circadian networks to reveal evidence for divergence between *B. rapa* paralogs that may be driven in part by variation in conserved non-coding sequences (CNS). Additionally, differential drought response among retained paralogous pairs suggests further functional diversification. These findings support the rapid expansion and divergence of the transcriptional network in a polyploid crop and offer a new approach for assessing paralog activity at the transcript level.

**\*For correspondence:**
greenham@umn.edu (KG);
c.robertson.mcclung@dartmouth.edu (CRMC)

[†]These authors contributed equally to this work

**Competing interests:** The authors declare that no competing interests exist.

## Introduction

The transition from basic research in Arabidopsis to new model systems for monocot and dicot crops has focused attention on the implications of polyploidy for current models of genetic processes developed in Arabidopsis. The expansion of gene content through whole genome duplication (WGD), tandem duplication, or transposed duplicates has been predicted to account for the evolution of morphological complexity (*Freeling and Thomas, 2006*). Improving crop yield in rapidly changing climates depends on our ability to integrate these gene content expansions into functional classifications of physiological importance. This will rely on the growing collection of sequenced genomes, not just across crop species but of ecotypes within species, including complementary genomic datasets such as transcriptomes, methylomes, chromatin accessibility profiles, and metabolomes. One difficulty in assigning new or overlapping functions among paralogs arises from heterogeneity in transcript abundance datasets generated under various environmental conditions, from various tissue types, and at distinct times of the day. Many studies have explored the potential for functional divergence of duplicated genes by comparing the expression levels normalized across a collection of expression studies (*Blanc and Wolfe, 2004*; *Ganko et al., 2007*; *Schnable et al., 2011*; *Woodhouse et al., 2014*) which limits the search to genes showing very dramatic differences in transcript abundance at a single time point.

**eLife digest** Like animals, plants have internal biological clocks that allow them to adapt to daily and yearly changes, such as day-night cycles or seasons turning. Unlike animals, however, plants cannot move when their environment becomes different, so they need to be able to weather these changes by adjusting which genes they switch on and off. To do this, plants keep track of how long days are using external cues such as light or temperature. One of the effects of climate change is that these cues become less reliable, making it harder for plants to adapt to their environment and survive.

This is a potential problem for crop species, like *Brassica rapa*. This plant has many edible forms, including Chinese cabbage, oilseed, pak choi, and turnip. It is also a close relative of the well-studied model plant, *Arabidopsis*. Since evolving away from *Arabidopsis*, the genome of *B. rapa* tripled, meaning it has one, two, or three copies of each gene. This has allowed the extra gene copies to mutate and adapt to different purposes. The question is, what impact has this genome expansion had on the plant's biological clock? One way to find out is to perform RNA-sequencing experiments, which record the genes a plant is using at any one time.

Here, Greenham, Sartor et al. report the results of a series of RNA-sequencing experiments performed every two hours across two days. Plants were first exposed to light-dark or temperature cycles and then samples were taken when the plants were in constant light and temperature. This revealed which genes *B. rapa* turned on and off in response to signals from the internal biological clock. It turns out that the biological clock of *B. rapa* controls close to three quarters of its genes. These genes showed distinct phases, increasing or decreasing in regular patterns. But the different copies of duplicated and triplicated genes did not necessarily all behave in the same way. Many of the copies had different rhythms, and some increased and decreased in patterns totally opposite to their counterparts. Not only did the daily patterns differ, but responses to stressors like drought were also altered. Comparing these patterns to the patterns seen in *Arabidopsis* revealed that often, one *B. rapa* gene behaved just like its *Arabidopsis* equivalent, while its copies had evolved new behaviors.

The different behaviors of the copies of each gene in *B. rapa* relative to its biological clock allow this plant to grow in different environments with varying temperatures and day lengths. Understanding how these adaptations work opens new avenues of research into how plants detect and respond to environmental signals. This could help to guide future work into targeting genes to improve crop growth and stress resilience.

The importance of daily rhythms was recognized with the 2017 Nobel prize in physiology or medicine to Jeffrey Hall, Michael Rosbash, and Mike Young for their discoveries of the molecular mechanisms generating circadian rhythms in *Drosophila* (*Rosbash, 2017*; *Young, 2018*). The conservation of circadian oscillators across the animal and plant lineages supports a role for these rhythms in maintaining fitness and evolving new regulatory pathways to fulfill that role (*Bell-Pedersen et al., 2005*). Many lines of evidence support the importance of circadian rhythms to plant biology, including photosynthesis, starch metabolism, biomass accumulation, and reproduction (*Greenham and McClung, 2015*; *Millar, 2016*). The circadian clock responds to environmental conditions to set these circadian rhythms to local time (*Greenham and McClung, 2015*). As a consequence, circadian rhythms and thus much of plant biology are likely to be influenced by climate change. Examples of natural variation in plant circadian function are accumulating, as is evidence that many domestication traits that facilitated the geographic expansion of crops are due to alterations in circadian clock genes (*Nakamichi, 2015*; *Müller et al., 2016*; *Müller et al., 2018*). This supports the utility in targeting circadian clock processes as a means of trait improvement without disrupting critical pathways required for growth and yield.

Plant circadian biologists have focused primarily on Arabidopsis as a model for defining circadian clock components and function in plants (*Creux and Harmer, 2019*). Transcriptome studies have revealed extensive circadian control of transcript abundance resulting in time of day changes in expression (*Covington and Harmer, 2007*; *Mockler et al., 2007*; *Michael et al., 2008*). These rhythmic changes in transcript abundance are not unexpected given the daily changes in light,

temperature, and precipitation that affect physiological processes such as photosynthesis. Dynamic changes in metabolism and physiology must be driven by dynamic changes in gene expression and ultimately protein regulation and activity. To identify candidate circadian regulators for trait improvement in crops, a more detailed time-course resolution of transcript abundance levels is needed to confirm whether the diel and circadian patterns observed in Arabidopsis are maintained in polyploid crops. The crop plant *Brassica rapa* offers an excellent model system for studies in crops. It is a member of the Brassicaceae and a close relative of Arabidopsis making comparative studies feasible. The morphological diversity in *B. rapa* with turnip, Chinese cabbage, pak choi, leafy and oil-type varieties provides a wealth of phenotypic traits to study in one species allowing for broad applicability to other crops. Preliminary studies have shown diversity in circadian clock parameters among morphotypes that correlate with various physiology measures suggesting that circadian clock variation has contributed to *B. rapa* diversification (*Yarkhunova et al., 2016*). Examination of the orthologs of known circadian clock genes in Arabidopsis revealed preferential retention of these genes in *B. rapa* following the triplication and extensive fractionation of the genome after the divergence of Brassica and Arabidopsis from their common ancestor around 24 million years ago (MYA; *Lou et al., 2012*). The preferential retention of clock genes is consistent with their involvement in protein complexes and regulation of critical pathways making them sensitive to dosage effects. The gene dosage balance hypothesis proposes that duplication of the entire genome is favored over single or chromosome level duplications because it maintains the appropriate concentration of gene products (*Conant et al., 2014*). This is supported by studies in yeast where genes of protein complexes tend to be lost simultaneously with their interacting proteins (*Pires and Conant, 2016*). The increase in expression of one duplicate could lead to or permit the loss of the other duplicate or neo-functionalization (*Pires and Conant, 2016*).

To assess the functional significance of the retention of circadian clock genes in *B. rapa* and look for possible examples of neo-functionalization, we performed two high-resolution circadian transcriptome experiments to characterize the circadian network. To compare the expression dynamics of paralogous genes, we developed a novel method for identifying and classifying changes in expression patterns, an R package called DiPALM (Differential Pattern Analysis via Linear Models). DiPALM facilitated a comparison of paralog expression patterns, revealing the genome-wide expansion of phase domains among paralogs providing novel insight into the rapid divergence of the transcriptional network in this crop. To identify transcriptional regulators exhibiting functional divergence, we compared the circadian gene regulatory networks (GRNs) between *B. rapa* and Arabidopsis. Using previously generated circadian microarray data in Arabidopsis, we compared GRNs to identify the more Arabidopsis-like versus the more divergent (less Arabidopsis-like) among members of *B. rapa* paralogous transcription factors (TF) gene pairs based on conservation of connected targets in the network. The identification of the more Arabidopsis-like TF ortholog was supported by the presence of conserved noncoding sequences (CNSs) surrounding TF target genes, reinforcing the importance of these CNSs for regulating gene expression. To associate these diverging patterns with a biological pathway, we applied DiPALM to our recent drought time-course experiment in *B. rapa* (*Greenham et al., 2017*) and discovered differential responses to mild drought stress among paralogs suggesting neo- and sub-functionalization.

## Results

### Circadian regulation of the *B. rapa* transcriptome is pervasive

Studies in Arabidopsis have demonstrated widespread circadian regulation of the transcriptome with distinct and overlapping genes responding to various entraining conditions (*Covington et al., 2008*; *Michael et al., 2008*). As noted above, genes contributing to circadian clock function have been retained in multiple copies following a WGD in *B. rapa* (*Lou et al., 2012*). We wondered whether there have been concomitant effects on the extent of circadian regulation of the *B. rapa* subsp. *trilocularis* (Yellow Sarson) R500 (henceforth called *B. rapa* R500) transcriptome and the expression patterns of circadian regulated paralogs. We conducted two RNA-seq experiments designed to capture the genes under circadian regulation entrained by light and temperature zeitgebers (German for 'time givers'). We entrained plants to light-dark (LD) cycles at constant temperature or thermocycles (HC) in constant light before transfer to constant light at 20℃ (LLHH; see

Materials and methods). Following 24 hr in constant conditions leaf tissue from the youngest leaf was collected and flash-frozen in liquid nitrogen every 2 hr for 48 hr. All samples were harvested from biological replicates of plants from the LD and HC entrainment that had been transferred to constant conditions of LLHH.

To identify the circadian transcriptome, we analyzed the LD and HC datasets and ran the circadian analysis program RAIN (*Thaben and Westermark, 2014*), a nonparametric method for the detection of rhythms from a variety of waveforms that are typical of transcript abundance datasets. The 2 hr sampling regimen provided the resolution to capture more cycling genes than possible with typical 4 hr sampling (*Hughes et al., 2017*). Using a Benjamini-Hochberg corrected p-value of 0.01, we identified 16,447 high confidence circadian regulated genes from the two datasets. Of the 22,204 genes that were expressed in the RNA-seq datasets, 74% of them passed our cutoff for cycling in one or both conditions, indicating retention of circadian regulation of the transcriptome following WGD in *B. rapa*. To assign cycling genes to specific phase bins based on the timing of peak expression, we generated co-expression networks for each dataset using the weighted gene correlation network analysis (WGCNA) package in R (*Langfelder and Horvath, 2008*), which has proven to be an effective method for grouping similarly phased genes based on their expression pattern (*Greenham et al., 2017*). This resulted in 14 modules in the LD dataset and 10 modules in the HC dataset. To demonstrate the uniformity of the genes within each module, a heatmap was generated with the log$_2$ transformed expression data of each gene for all modules numbered based on their phase (*Figure 1*). Each module reflects a similarly phased set of genes that collectively are phased throughout the day with the LD_01 module showing peak expression at ZT24 (subjective dawn; ZT refers to Zeitgeber Time and indicates the number of hours since the last dark-to-light or cold-to-warm transition) and LD_09 showing peak expression at ZT36 (subjective dusk).

The similarity in patterns seen in the LD and HC heatmaps in terms of phasing and distribution of genes within those phase bins suggests that there may be considerable overlap in gene phasing

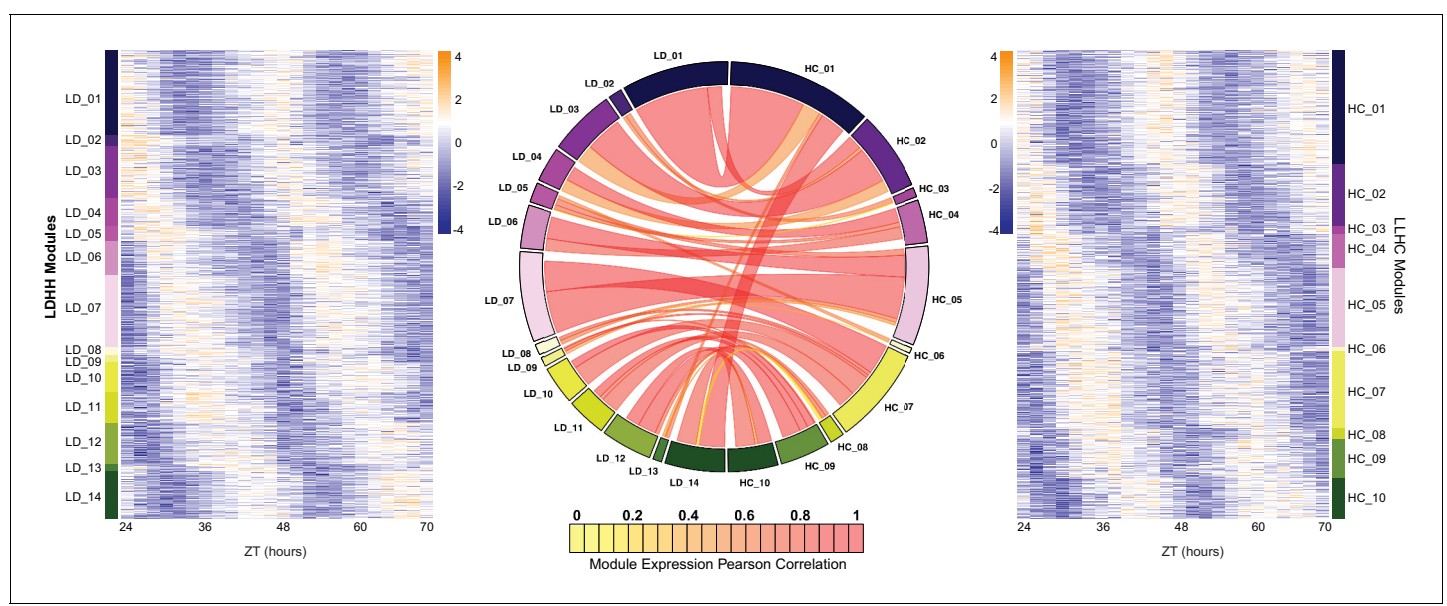

**Figure 1.** Expansion of the circadian transcriptome in *B. rapa*. Co-expression networks were generated for the LD and HC datasets (named for the entraining conditions; all samples were from plants transferred to constant light and temperature [LLHH]). The 14 modules from the LD dataset are shown in the heatmap on the left and the 10 modules from the HC dataset are shown on the right. Heatmaps were generated using the log$_2$ transformed FKPM expression data. The low expression level is in purple and the high expression level is in orange. The circos plot in the middle shows the overlap in genes between the LD and HC modules. Modules are numbered based on their phase starting at the beginning of the day (ZT24). The width of the ribbon signifies the number of genes in common between the connected modules and the color represents the Pearson correlation coefficient between the eigengenes of the two datasets. The high correlations (median of 0.73) indicates no or little phase difference between LD and HC entrainment for most genes.

The online version of this article includes the following figure supplement(s) for figure 1:

**Figure supplement 1.** Phase difference of genes between LD and HC entrainment.

after LD and HC entrainment. To quantify the overlap, we matched the genes across the two networks and compared the correlation of eigengenes (first principle component of the expression matrix for each module) between LD and HC modules. The circos graph in *Figure 1* depicts the overlap between the two networks where the width of the ribbon represents the number of genes in common between the two modules and the color signifies the Pearson correlation coefficient between the eigengenes of the two datasets with dark orange being a correlation coefficient of 1. Because the modules are numbered based on their phase, similarly phased modules are arranged in the same order in the circos plot and the significant overlap and expression pattern between these modules is evident. *Figure 1—figure supplement 1* shows the distributions of phase difference between LD and HC for all 16,447 genes that cycle in both conditions. 13,850 genes (~84%) have a phase difference of less than or equal to 4 hr. These comparisons demonstrate that most genes have the same or similar phasing when entrained by either photocycles or thermocycles. To further assess the similarity between these two datasets, GO ontologies were compared for each module from the LD and HC experiments (*Supplementary file 1*). This revealed similar biological processes enriched in the modules with a high correlation in expression patterns. For example, genes in LD_03 and HC_02 were both significantly enriched (FDR adjusted p-value<0.01) for photosynthetic processes and response to abiotic stress, consistent with their morning phased expression. Interestingly, genes in LD_07 and HC_07 were significantly enriched for protein phosphorylation suggesting a time-of-day regulation of this process, as in Arabidopsis (e.g. *Choudhary et al., 2015*).

The correlation between module membership is not a rigorous test of differential transcript abundance and many genes have low correlations with their module eigengene that are not reflected in the analysis in *Figure 1*. To the best of our knowledge, there was not a rigorous test available for identifying significantly different gene expression patterns that would classify the change based on phase, amplitude, or a combination of both. Rather than looking at differentially expressed genes at any given time point, we felt it was more important to classify a pattern change that encapsulated the entire time course. Therefore, we developed an R package, DiPALM, which takes advantage of the network analysis that assigns an eigengene to each module and thereby produces a minimal set of patterns representing the entire dataset. The expression correlation of a given gene to any module's eigengene defines the module membership (kME) of that gene to the module. The combination of kMEs for a gene across all modules can be used to encode its expression pattern numerically and allows for quantitative comparisons between any two gene's expression patterns. This allowed us to run a set of linear model contrasts (one for each eigengene) that is analogous to running a contrast of gene expression data between time points or treatments except in this case the kME value represents the entire expression profile across the time course. We first tested for differential expression patterns between the LD and HC datasets for the set of 15,101 genes that cycle after entrainment to either LD and HC. Both datasets were collected in constant conditions of light and temperature (LLHH) following different entrainment regimens as specified by their names. To generate a significance cutoff, we also ran the analysis on a permuted gene expression set where gene accessions were randomly re-assigned to expression patterns. P-values were then calculated using this permuted set. Using a p-value cutoff of 0.01, we identified just 1713 genes, or 11% of all cycling genes, that have altered patterns following LD versus HC entrainment. To quantify overall expression level variation, we ran a similar linear model analysis on the median expression level for all genes and identified 3465 (23%) of genes, only 448 of which overlapped with the pattern change list (*Supplementary file 2*). The 11% of cycling genes with entrainment-dependent cycling patterns are interesting but not within the scope of this manuscript. A functional enrichment analysis of these genes revealed translation initiation factors and ncRNA metabolic process among the significant (FDR adjusted p-value<0.01) functional categories (*Supplementary file 1*, 'Differential_Pattern' Tab). For the remainder of analyses with these datasets, we have combined LD and HC to increase our statistical power by having four replicates per time point rather than two. One significant advantage of a linear model-based framework is the ability to account for any identified effect. We make use of this feature by modeling any LD versus HC entrainment effects as a covariate. In other words, DiPALM gives us the ability to combine these datasets while still accounting for any differences between LD and HC entrainment. This combined dataset (referred to as LDHC) provides more statistical power for further analysis, particularly for the 89% of genes that show no significant change between LD and HC. For the other 11% that do have differences, these differences are taken into account and will not adversely affect the results.

## Circadian regulated genes retained in multiple copies exhibit diverged expression phasing

The network analysis revealed that a large portion of the transcriptome exhibits rhythmic expression patterns. This implies that multi-copy paralogs have retained their rhythmic expression patterns and circadian regulation consistent with the preferential retention of circadian clock orthologs in *B. rapa* (*Lou et al., 2012*). Based on the gene dosage model, the balance in expression among different subunits of protein complexes must be maintained resulting in the proper adjustment of paralog expression level, in some cases resulting in one copy maintaining high expression while the other is repressed (*Conant et al., 2014*). To evaluate this model, we calculated the mean expression level for all cycling genes across the 48 hr time course. The mean expression level for the set of retained multi-copy paralogs was significantly higher (p-value=0.0 based on one-way ANOVA with Tukey's test) than for genes retained in a single copy (*Figure 2A*). It is possible that one of the retained copies is expressed at a much higher level than the average single-copy gene as well as its paralog. To test whether this is the case, we separated all the cycling two- and three-copy paralogs into the highest and lowest expressed copies based on the average expression level across all time points. In the case of three-copy paralogs, we only included the highest and lowest expressed genes in the analysis. We compare this to randomly paired single-copy gene pairs that were also separated into high and low expression groups (*Figure 2B*). Surprisingly, the difference is observed in the low expressed paralog where these duplicated genes have significantly higher (p-value=0.0 based on one-way ANOVA with Tukey's test) average expression levels compared to the genes retained in a single copy, suggesting that although the duplicate paralogs do appear to exhibit gene dosage, their overall expression is retained at a higher level than the expression of single-copy genes.

The retention of multi-copy genes that are under circadian regulation and maintained at a relatively high expression level led us to explore whether there is evidence for divergence in expression pattern that would support neo- or sub-functionalization among paralogs. To associate similar patterns, we applied the same WGCNA method to the combined LDHC dataset as was done for the individual analyses shown in *Figure 1*. This resulted in 12 modules with distinct phasing throughout the day that is visible when the eigengene expression for each module is presented as a heatmap (*Figure 2C*). Next, we wondered whether there was any association between the phase of expression and retained copies that may suggest certain biological processes that are phased to specific times of the day and may preferentially retain multi-copy genes. Based on the number of genes within each module, we ran a hypergeometric test to look for over- and under-enrichment of multi-copy genes within the modules. Surprisingly, we found that modules with phasing from morning to midday tend to be enriched for multi-copy genes. By contrast, evening and night phased modules were depleted for multi-copy genes (*Figure 2D*). These trends were not associated with the number of genes within the module as can be seen with two of the largest modules, LDHC_02 and LDHC_09, being over- and under-enriched, respectively. GO enrichment was carried out on a combined group of all multi-copy genes from the five morning modules with significant enrichment for multi-copy genes (p-value<0.05). The same was done for the group of all multi-copy genes from all four evening modules with significant depletion in multi-copy genes. Both of these sets appear to be representative of the whole modules from which they came with the morning-phase copied genes being significantly enriched (FDR adjusted p-value<0.01) for photosynthesis, translation, and response to abiotic stimulus genes. The evening-phase multi-copy genes were significantly enriched for protein phosphorylation and glycosinolate biosynthesis genes (*Figure 2D*, *Supplementary file 3*).

To look for signs of possible neo- or sub-functionalization among paralogs, we compared expression profiles to identify paralogs with significantly different expression patterns. For three-copy paralogs, where all three copies (a, b, and c) were expressed, these sets were converted into three, two-member pairs (ab, bc, and ac). We applied a similar linear modeling test using DiPALM that we ran on the LD and HC comparison but including the covariate in the model to account for differences following LD versus HC entrainment. We ran this analysis on 4433 pairs where both genes are expressed. We found 3743 (84%) pairs with differential median expression and 1883 (42%) pairs exhibiting differential expression patterns (p-value<0.01), the vast majority (1607/1883; 85%) of which showed both differential median expression and pattern (*Supplementary file 4*). However, this does not describe how the patterns differ. As with standard differential expression tests, it is

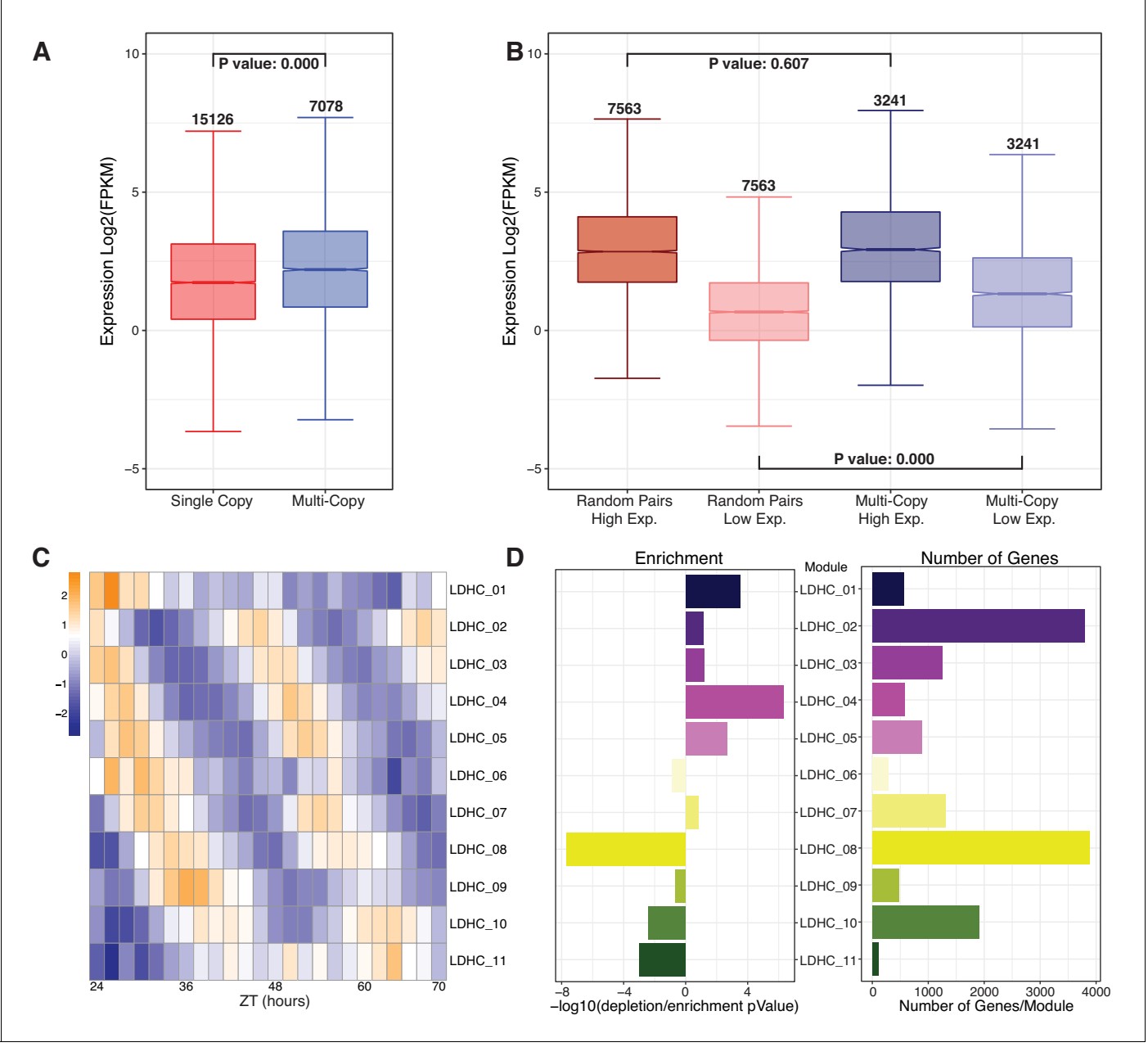

**Figure 2.** Retained multi-copy circadian regulated genes are highly expressed and display time of day variation. (**A**) Mean log₂ FPKM expression levels for each gene across the combined LD and HC time course for multi-copy paralogs compared to single-copy genes. Numbers above the whiskers indicate the number of genes in each group. P-value is the result of a one-way ANOVA with Tukey's test. (**B**) Expression level comparison when paralogs are separated into high and low expression groups compared to randomly paired single-copy genes. Numbers above the whiskers indicate the number of genes in the groups. P-value is the result of an ANOVA test. (**C**) Heatmap of the 11 modules of the combined LDHC co-expression network arranged by ZT time (in Zeitgeber [ZT] time, where ZT0 represents the most recent dark to light or cold to warm transition) across the x-axis and circadian phase along the y-axis. (**D**) Results of a hypergeometric test of the number of multi-copy genes in each module. The left barplot shows the results of the hypergeometric test expressed as a -log₁₀ pValue with enrichment to the right and depletion to the left of 0. The right barplot shows the number of multi-copy genes in each of the modules.

critical to associate a direction of change in expression to know how a gene transcript is affected by treatment or condition. To isolate the type of pattern change for the paralogous pairs exhibiting significantly different patterns, we performed clustering on the vector of expression values across the combined LDHC data for each gene whose expression differed significantly from its paralog. This clustering had the effect of grouping genes based on their phase. Similar clustering was done for

the significantly different median expression set. This clustering component is also part of the DiPALM package. A detailed description and example dataset of the analysis pipeline is provided with the package on CRAN (*R Development Core Team, 2018*).

To visualize the degree of pattern change, we generated a heatmap of each of the clustering methods with the paralogous gene pairs in the same row for comparison. As expected, the pattern clustering uncovered the changes in rhythmic patterns between pairs (*Figure 3A*) while the median expression clustering uncovered overall changes in transcript abundance (*Figure 3—figure supplement 1*). From the heatmap visualization, it is clear that the majority of the pattern change among paralogous pairs is the result of an altered phase of peak expression. In some cases (*Figure 3A and B*, clusters 4 and 12) the pairs are completely antiphase. This suggests a genome-wide expansion of phase domains among retained paralogs.

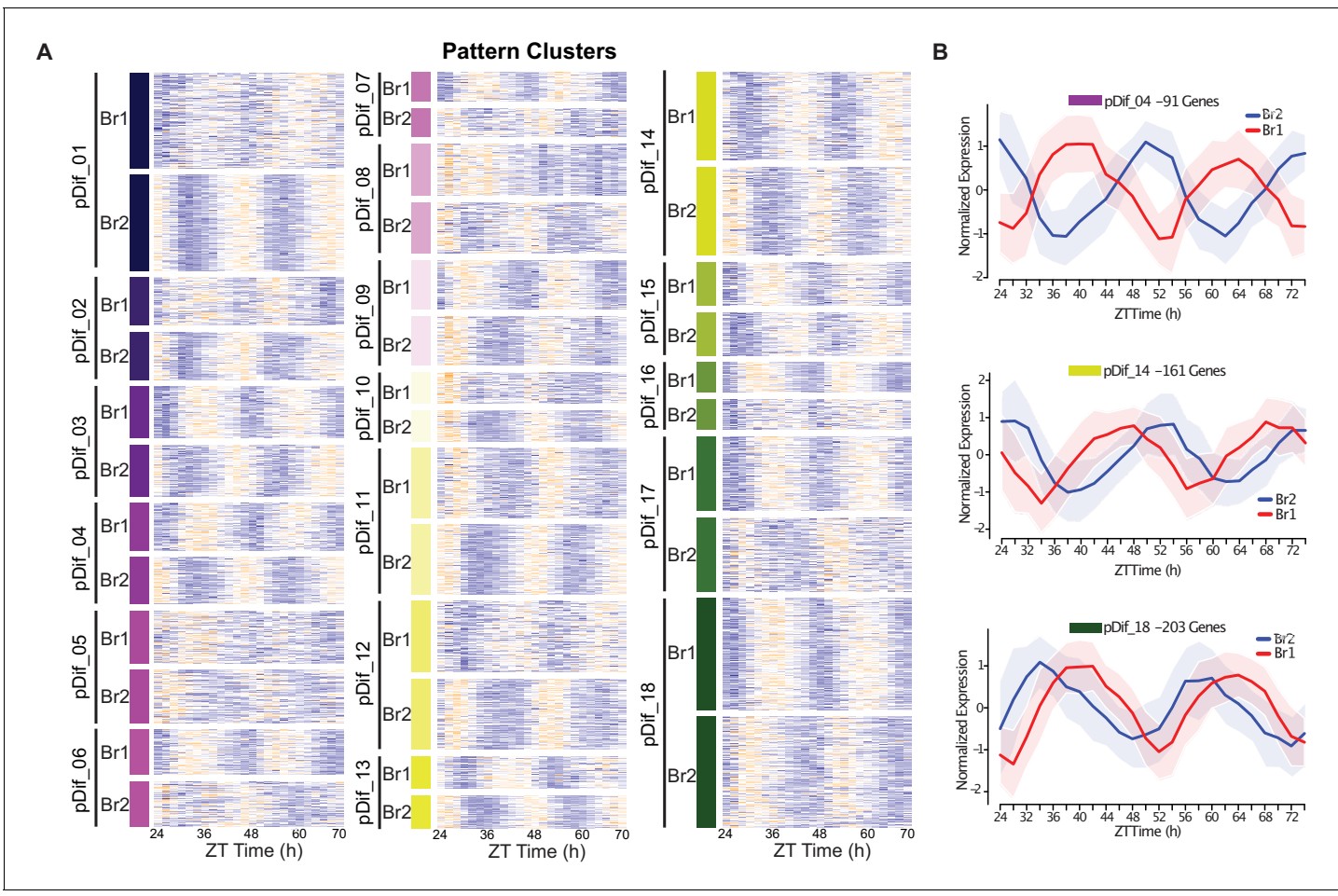

**Figure 3.** Divergence in phase domains among retained paralogs. (**A**) Heatmap of the results from the DiPALM differential pattern analysis (pDif) clustering showing the changes in expression pattern for paralogous pairs. The Br1 and Br2 heatmap blocks for each module contain matching paralogous pairs; for example, the gene in the first line of Br1 and Br2 heatmaps for module pDif_01 are paralogs. The blocks have been stacked on top of each other to be able to compare the change in pattern across the time course. Three-copy paralogs were split into three two-way comparisons. Expression values are log_2 transformed FPKM values and the expression is arranged by ZT time across the x-axis. Higher expression levels are in orange and low expression levels are in purple. For example, pDif_01 shows paralogous pairs that are anti-phase with Br1 paralogs peaking at ZT36 and ZT60 and Br2 paralogs peaking at ZT24 and ZT48. (**B**) Line plots showing the expression patterns for three modules (pDif_04, pDif_14, and pDif_18). Each plot shows the normalized expression of paralogous pairs for all the genes in the module. Ribbons (shaded regions) represent the standard deviation. The online version of this article includes the following figure supplement(s) for figure 3:

**Figure supplement 1.** Divergence in median expression levels among retained paralogs.

## Identifying the 'Arabidopsis-like' paralog of *B. rapa* using gene regulatory networks

The divergence in expression pattern among retained paralogs led us to ask how diverged the retained pairs are with respect to their Arabidopsis ortholog (*Figure 4A*) and whether one copy exhibits an Arabidopsis-like expression pattern while the other copy exhibits a different expression pattern. One method of comparing the orthologs between *B. rapa* and Arabidopsis is to compare the phase of expression. However, assigning an accurate phase to circadian data from two cycles is challenging and often gene expression patterns show very broad peaks in abundance that can be difficult to classify, especially with the resolution of only 4 hr that is available for Arabidopsis. Also, because we are comparing two species, we have to consider the properties of the clock in each species. The *B. rapa* R500 clock has a slightly shorter period than Arabidopsis Col-0 that results in altered phasing among genes that is not indicative of divergence in function but arises merely from the different paces of the Arabidopsis and *B. rapa* oscillators. For example, if we plot the expression of single-copy circadian clock genes from *B. rapa* and their corresponding orthologs in Arabidopsis we see a leading phase in *B. rapa* (*Figure 4—figure supplement 1*). To avoid these phase complications, we chose a gene regulatory network (GRN) approach using GENIE3 (*Huynh-Thu et al., 2010*) that would provide additional statistical robustness by first predicting transcription factor (TF) targets based on expression dynamics within each species followed by a comparison of network connections between the species (*Figure 4B*).

We obtained previously published circadian microarray data from Arabidopsis that were generated under similar conditions with LD and HC entrainment (LL_LDHC, LL_LLHC, LL12_LDHH, and LL23_LDHH from *Mockler et al., 2007*). We selected Arabidopsis TFs from the Arabidopsis TF database (https://agris-knowledgebase.org/AtTFDB/) and *B. rapa* TFs from the Mapman annotation 'RNA regulation of transcription' in addition to known circadian clock TFs. This resulted in a list of 612 Arabidopsis and 2147 *B. rapa* TFs that were expressed in their respective datasets. For the target set, we included 9201 Arabidopsis expressed genes and the corresponding 14,541 *B. rapa* expressed orthologs. Separate GRNs were generated for Arabidopsis and *B. rapa*. To identify the significance of TF-target edges, we generated a permuted network by using the same TFs but shuffling the expression values for the target genes. These permuted network edges were used as a null distribution to select edges from the actual networks with a 5% FDR, resulting in an Arabidopsis GRN with 71,216 edges and a *B. rapa* GRN with 947,062 edges. A gene was said to be a target of a TF if one of these significant edges existed between them. We hypothesized that the paralogous TF in *B. rapa* that displayed more of the Arabidopsis orthologous function would have a greater overlap in targets in the network compared to the more divergent pair member. In other words, using TFs as features to describe the targets, how well can the expression of the target genes be explained by the expression of that TF (*Figure 4B*)? A set of 256 TFs exists where one Arabidopsis TF can be associated with two *B. rapa* paralogs and all three of these genes had target groups defined by their respective GRNs.

For each of these 256 sets, we examined the significance of the overlap between the Arabidopsis TF target group versus the corresponding *B. rapa* orthologous TF target groups. This resulted in two p-values for each group indicating how similar each *B. rapa* TF is to its orthologous Arabidopsis TF in terms of target gene overlap (*Supplementary file 5*). Next, we wanted to determine if the difference in these two p-values was significant; that is, does one of the *B. rapa* TFs show more conservation of target gene overlap with its Arabidopsis ortholog than the *B. rapa* paralogous TF? This was accomplished with another permutation-based test where genes were randomly sampled to create target groups of the same sizes. The two *B. rapa* versus Arabidopsis p-values were calculated and the difference was taken. This was repeated 10,000 times for each of the 256 TF sets. 49 TF pairs exhibited significant enrichment (p-value<0.05) for one *B. rapa* TF (assigned Br1) being more similar to Arabidopsis than its paralog (*Supplementary file 5*), suggesting possible divergence in function between these TFs (*Figure 4C and D*). It is worth noting that the size of the target group is not driving the enrichment as we see a broad distribution in target size and significance (*Figure 4C*). Among the list of 49 TFs, six are part of the core circadian clock (*ELF3, ELF4, PRR9, PRR7, PRR5,* and *TOC1*), and a seventh, *RVE1*, integrates the circadian clock and auxin pathways (*Rawat et al., 2009*; *McClung, 2019*). Based on the predicted targets of these TFs in the GENIE3 model, there are 11,559 *B. rapa* and 3387 Arabidopsis genes regulated by these 49 TFs, providing further support for

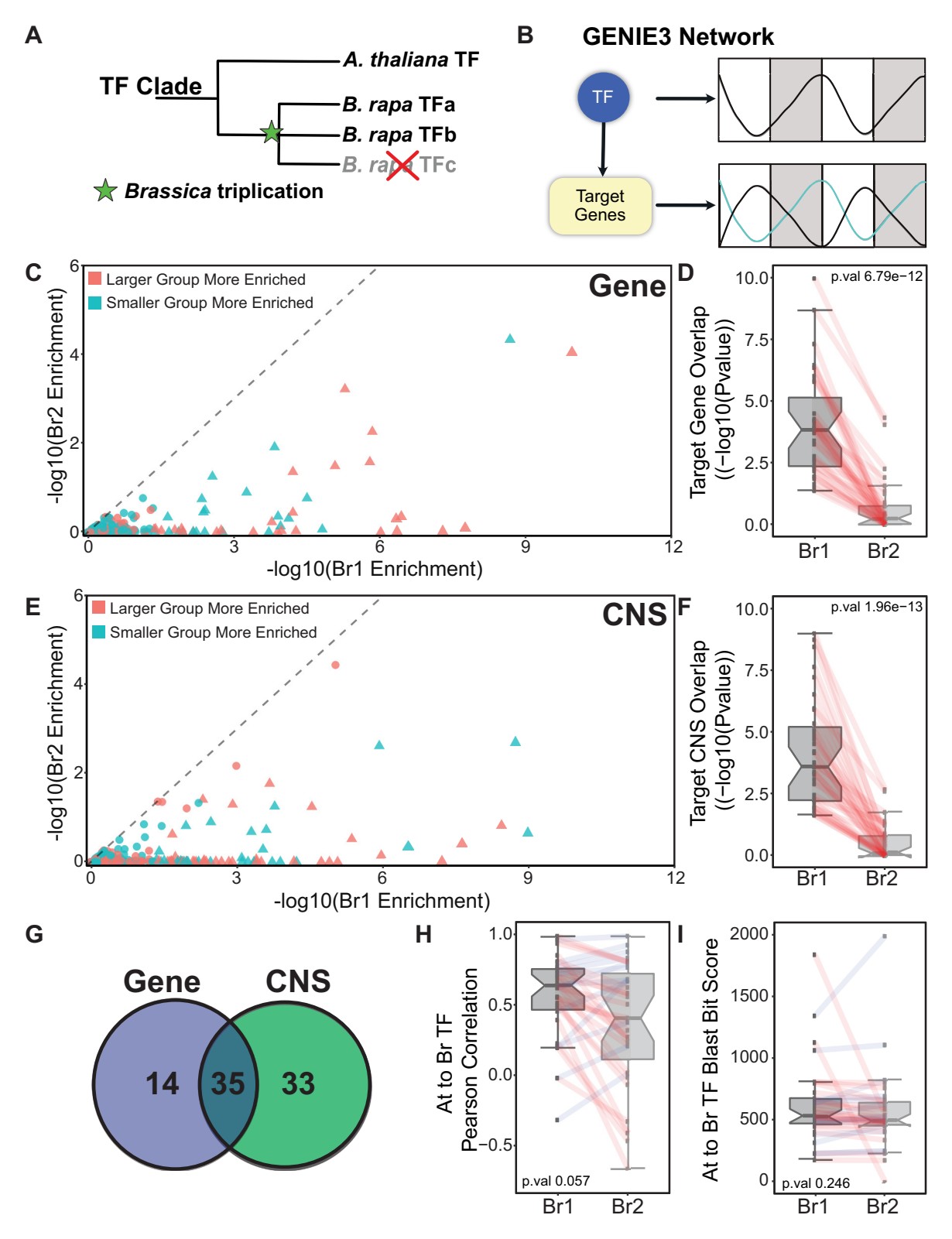

**Figure 4.** Identifying the 'Arabidopsis-like' paralog using GRNs. (A) Schematic showing the triplication event following the divergence between Arabidopsis and *B. rapa* leading to multi-copy orthologs of known Arabidopsis transcription factors (TF). (B) GENIE3 networks are generated to associate a TF with target genes based on the expression patterns of the TF and all the genes in the network. There is no assigned direction to the TF regulation in our GENIE3 network resulting in possible positive or negative regulation. (C) Scatterplot showing the results for the gene overlap network

*Figure 4 continued on next page*

*Figure 4 continued*

analysis for all 256 *B. rapa* TF pairs. Triangles indicate TF pairs that were identified as having one *B. rapa* paralog significantly more Arabidopsis-like (p-value<0.05, based on target group overlap) than the other; circles were not significantly different. Points colored red means that Br1 (the more Arabidopsis-like TF) has a larger target group and teal means Br1 has the smaller target group. (D) P-value distribution of the overlap in the *B. rapa* versus Arabidopsis target genes from the analysis shown in panel (C). All Br1 and Br2 TFs are viewed as separate groups with paralogs connected by red lines. (E) Same scatterplot presentation as panel (C) showing the results from the CNS overlap analysis. (F) P-value distributions of the overlap in *B. rapa* versus Arabidopsis target CNSs from the analysis shown in panel (E). (G) Of the 49 TFs identified as more Arabidopsis-like based on target genes between the Arabidopsis and *B. rapa* networks, 35 overlapped with the 68 TFs identified based on CNSs between Arabidopsis and *B. rapa* target groups. (H) Distribution of the Pearson correlation of TF expression patterns comparing each Br1 and Br2 TF with Arabidopsis TF ortholog. (I) Distribution of BLAST bit score of Br1 and Br2 TFs compared to their orthologous Arabidopsis TF.

The online version of this article includes the following figure supplement(s) for figure 4:

**Figure supplement 1.** Phase variation between Arabidopsis and *B. rapa* clock genes.
**Figure supplement 2.** Motif enrichment in CNS regions in Arabidopsis and *B. rapa* GRN TF target genes.

an expansion of the circadian network in *B. rapa*. The divergence in TF target genes indicates several possible changes have occurred; these could include modifications to regulatory elements of the target genes, mutations that alter the TF protein binding efficiencies for motifs or interacting partners, or a combination of both. Alterations to regulatory elements associated with core TFs and/or target genes can lead to whole pathway-level restructuring. One possible mechanism for altered expression regulation is the distribution of conserved noncoding sequences (CNSs). A set of CNSs was identified across the Brassicaceae that show signs of selection (*Haudry et al., 2013*). To associate CNSs with the *B. rapa* R500 genome, we performed a BLAST analysis with a collection of ~63,000 CNSs against the *B. rapa* R500 genome. Provided the alignment met our BLAST filters, we allowed each CNS to have a maximum of three targets (see Materials and methods). We repeated the BLAST with the Arabidopsis genome but restricted each CNS to one target gene.

To test for altered regulatory element occurrences between target genes of the identified diverged TFs, we asked whether variation in CNS retention followed the same pattern as the observed gene expression changes. Do we see a similar divergence in *B. rapa* paralogous TF enrichment with Arabidopsis in the GENIE3 network if we replace the target set gene ortholog data with CNSs? With the list of genes and associated CNSs, we replaced the target genes in the GENIE3 networks with CNSs resulting in a network with TFs targeting a group of CNSs rather than genes. We performed the same permutation tests to assign significant enrichment to the groups to ask whether we could identify a *B. rapa* R500 TF ortholog that was more Arabidopsis-like than its paralog. We identified 68 significant TFs (p-value<0.05) in the CNS network (*Figure 4E and F*, *Supplementary file 5*), 35 of which overlapped with the 49 TFs identified based on target gene overlap (*Figure 4G*). The agreement between these two approaches is apparent when the corresponding p-value distributions from the overlap of targets are plotted (*Figure 4D and F*). In these boxplots, the Br1 enrichment for At target gene overlap is shown with the paralogous pairs connected by the red lines. Not only does the agreement between the target gene and CNS overlap further strengthen the support for those 35 TF pairs showing signs of divergence but suggests that the CNS distribution is associated with gene expression patterns and is a good predictor of expression variation. With this set of high confidence diverged TFs from the overlap group, we wondered whether changes in TF amino acid sequence contribute to the divergence between paralogous pairs in which case we would expect the Arabidopsis-like *B. rapa* TF to be more similar than its paralog. To test this, we ran a protein BLAST using the *B. rapa* TFs against the Arabidopsis genome and examined the distribution of blast scores for the more and less Arabidopsis-like TF. Results from the BLAST suggest very little association between amino acid sequence and TF divergence (*Figure 4I*) suggesting that changes to regulatory regions associated with target genes are likely to be a major driver of TF divergence. TF expression pattern changes are likely contributing to the divergence in regulation. To test this further, we conducted a similar analysis to the BLAST comparison where we looked at the expression correlation of *B. rapa* TFs versus the orthologous Arabidopsis TF. In general, the more Arabidopsis-like *B. rapa* TF was more likely to maintain a higher expression correlation to the Arabidopsis ortholog but the effect is not significant (*Figure 4H*, p-value 0.057) and several of the 35 TF sets tested do not show this result. However, due to the period difference between the Arabidopsis and *B. rapa* datasets (longer period confers delayed phase), it is difficult to compare

expression patterns directly (*Figure 4—figure supplement 1*). While paralogous TFs likely bind to the same motifs, a divergence in expression may result in the loss of transcriptional coactivators or corepressors required for gene activation or repression due to temporal separation in expression pattern resulting from the shift in phase of that TF. Similarly, an altered phase of expression might allow interaction with new TF interacting factors to provide a new target affinity for that TF resulting in a new target set.

If the CNSs are driving the target gene expression differences we would expect them to be enriched for TF binding motifs (*Haudry et al., 2013*). To test for motif enrichment, we selected 12 of the TF groups where Br1 was significantly more Arabidopsis-like (see *Supplementary file 5*, third tab). For each TF (*B. rapa* paralogs and Arabidopsis ortholog), we took the collection of CNSs represented by their target genes in the GRNs and ran them against the HOMER motif analysis algorithm (*Heinz et al., 2010*). Because the CNSs are located throughout the gene (promoter, 5'UTR, introns, 3'UTR) we selected the sequence from 2 kb upstream of the start codon to the 3'UTR for each target gene for comparison. We also included just the 2 kb upstream sequence to compare to standard motif search parameters. For all 12 TFs tested, we found three– to fivefold greater enrichment for motifs in the CNS elements compared to the full length and 2 kb promoter background sets (*Figure 4—figure supplement 2*). This is consistent with the results from the GRN analysis showing that CNSs are as predictive as expression dynamics and contain important regulatory elements. Further studies are needed to look for associations between groups of CNSs with their corresponding binding motifs and specific gene expression patterns. Since the GENIE3 algorithm associates target genes based on a TF being an activator or repressor, the target genes typically have several major expression patterns (*Figure 4B*). Isolating distinct patterns and analyzing CNS variation between the target groups may reveal new motif groupings or novel motifs.

## *B. rapa* paralog expression pattern response to abiotic stress

The gene balance hypothesis posits that multi-subunit complexes are sensitive to variations in stoichiometry resulting in dosage compensation to produce the same amount of product (*Birchler and Veitia, 2014*). Clustering based on paralog median expression levels did reveal a consistent trend with one paralog having significantly higher median expression levels compared to the other retained paralog of that pair (*Figure 3—figure supplement 1*). However, as previously demonstrated, the overall expression levels for multi-copy genes is higher than single-copy genes (*Figure 2A and B*). In addition, the rhythmicity in the paralogous pairs is still apparent, providing further support for focusing on the importance of the pattern of expression rather than simply the overall expression levels. This led us to wonder whether there is any indication that these pattern changes may contribute to new temporal responses to environmental stimuli such as abiotic stress. Gated stress response has been characterized in several plant species including Arabidopsis, poplar, rice, and *B. rapa* (*Fowler et al., 2005*; *Wilkins et al., 2009*; *Wilkins et al., 2010*; *Greenham et al., 2017*; *Grinevich et al., 2019*). If a time-of-day dependent stress-responsive gene in Arabidopsis is now present in two copies in *B. rapa* with altered expression patterns does this confer an expanded stress response window or does one copy exhibit stress response while the other does not?

To look for indications of divergence in the function, we used a mild drought time-course RNA-seq dataset (*Greenham et al., 2017*) to test for altered responses to drought among pairs of paralogs. We first used the well-watered control samples to identify the paralogous pairs with altered patterns. Consistent with the divergence in pattern change under circadian conditions, the same trend is apparent under the diel (LD) conditions of our drought time course. Out of 4664 total pairs where two copies show detectable expression levels, 3259 pairs had significantly different patterns under control conditions (p-value<0.01). In the circadian dataset, we observed just 42% of genes with altered pattern but these diel data reveal 70% of pairs with altered pattern. Similarly, 77% of pairs (3602) had significantly different median expression levels (*Supplementary file 6*). Of the total pairs, 50% had both genes identified as circadian-regulated and an additional 35% had one circadian-regulated gene, emphasizing the importance of the circadian clock in diel conditions. For the pairs with altered patterns under the well-watered conditions, 93% were tested (both genes were expressed) under circadian conditions and 52% showed significant pattern changes. These results point to even greater divergence of paralog expression patterns under real-world, diel conditions compared to circadian conditions.

To identify differential response to drought among paralogous pairs, we first ran DiPALM on all (23,248) expressed genes and tested each gene for pattern and expression changes in response to drought. We identified 3891 with a significant pattern change and only 327 with median expression changes (p-value<0.01; *Supplementary file 7*). Unlike the circadian dataset, the largest source of variation in expression is due to a phase change consistent with the importance of time-of-day gating of the stress response (*Wilkins et al., 2010*; *Greenham et al., 2017*). This provides further support for the dynamic nature of expression regulation and the limitations of simply quantifying transcript abundance differences at single time points. It should also be noted that this drought treatment captured the early signs of drought perception with very subtle expression changes during the first 24 hr and more evident changes in the subsequent 24 hr (*Greenham et al., 2017*). The ability to detect these distinct patterns using DiPALM highlights the effectiveness of the network pattern approach for capturing unique and unpredicted patterns. To test whether the paralogous pairs exhibiting differential expression patterns under well-watered conditions are enriched for drought-responsive genes, we performed a permutation test. We randomly sampled the same number of pairs from the full set of pairs (3259 out of 4664) for 10,000 permutations to identify the likelihood of selecting drought-responsive genes within the 3891 sampled set. As a result, the differentially patterned copies were enriched for genes with drought responsive patterns (p-value 0.0005; *Figure 5A*), whereas copies with differential median expression level were not (p-value 0.6457; *Figure 5A*). With enrichment of drought-responsive genes among these pairs exhibiting different patterns, we wondered whether these pairs are more or less likely to have one or both copies

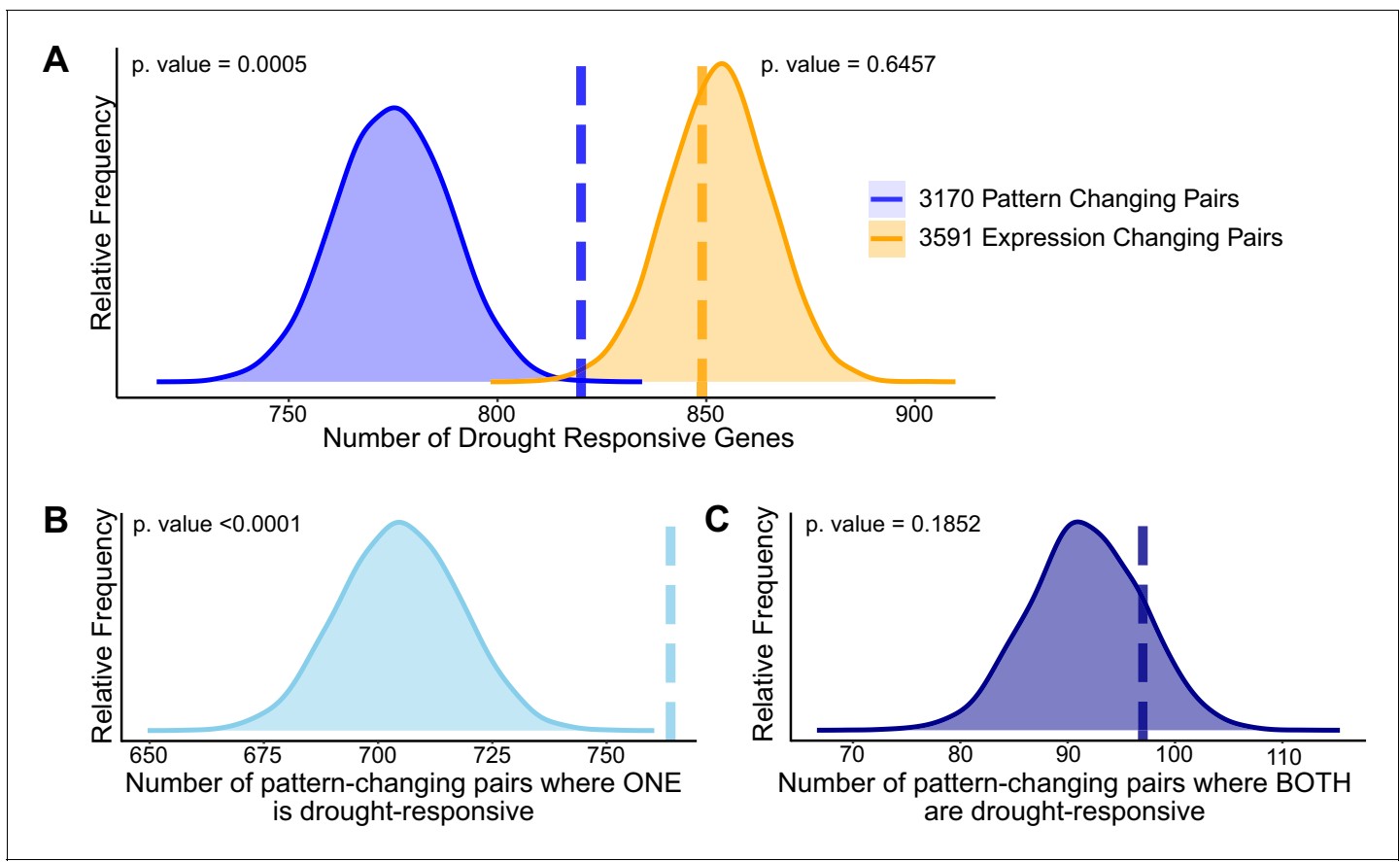

**Figure 5.** Divergence in drought responsiveness among retained paralogs. (A) Frequency distributions showing the results of a permutation test of the likelihood of paralogous pairs with significantly diverged expression patterns (blue) or significantly different expression levels (orange) in control conditions being drought responsive. Frequency distributions showing permutation results testing the significance of only one member of the paralogous pair being drought-responsive (B) or both paralogs being drought-responsive genes (C). The dashed vertical lines represent the true test statistic and the frequency distributions represent the null distributions that result from the 10,000 permutation tests. Supplementary information accompanies this paper.

responding to drought. Using the same 10,000 randomly sampled sets of 3891 pairs, we estimated a null distribution of the expected number of pairs with one and two drought-responsive genes. Results from the permutations indicated significant enrichment for pairs in which one member is drought responsive (p-value<0.0001) and no enrichment for both copies being drought responsive (p-value 0.1852; *Figure 5B and C*). Thus, we observed enrichment for only one but not both paralogs responding to drought stress in *B. rapa*. This suggests that the genome-wide expansion of expression domains among paralogs is biologically meaningful, in this case for drought stress response. More broadly, these results have important implications for how we capture and characterize transcriptomic responses or 'states' when making predictions about paralog function. Temporal, spatial, and conditional regulation can reveal new expression dynamics.

An assessment of functional comparisons among paralog expression levels defines the gene with the highest expression in one or multiple tissue samples as 'winning' over the other (*Woodhouse et al., 2014*). This classification is often referred to when looking for signs of subgenome dominance within polyploid species. With our set of pairs with one drought-responsive paralog, we wondered whether the responsive member of a pair had a higher median expression level under control conditions. Of the 764 pairs with a drought-responsive paralog, the drought responsive gene was the lower expressing member in 420 pairs and the higher expressing member in 344 pairs in the control conditions. Thus, we conclude that transcript abundance, whether at a single time point or combined across a time series, is not a reliable predictor of a gene's functional importance. As validation of this, we ran a standard linear model test at each time point comparing well-watered and drought treatments and did not detect significant transcript abundance differences for any genes. To detect the initial transcriptional response to drought perception, we had to incorporate the complete transcript profile into the differential expression test. What appears to be very subtle changes in the abundance of a subset of transcripts are contributing to the measured temporal physiological changes in Fv'/Fm' and stomatal conductance also occurring at specific time points (*Greenham et al., 2017*). The ability to capture these early transcriptomic responses to the onset of drought offers new insight into how responsive the network is to slight adjustments in the temporal regulation of expression. The next challenge is to capture this fine-scale resolution across genotypes with diverse physiological responses to stress and identify the associated patterns.

## Discussion

The circadian regulation of the transcriptome leads to time of day changes in gene expression that coordinate physiological responses to environmental conditions. This study emphasizes the power of *B. rapa* as a model system for investigating the consequences of polyploidy on transcriptional network dynamics. The close relationship of *B. rapa* to Arabidopsis facilitates comparative studies and guides gene function hypotheses due to the wealth of genomic and molecular resources developed in Arabidopsis. *B. rapa*, a morphologically diverse crop, has undergone whole-genome triplication since diverging from its common ancestor with Arabidopsis resulting in an expansion of gene copy number. To examine how this expansion has influenced the circadian transcriptome, we developed a new R package (DiPALM) to compare gene expression in time-course experiments. DiPALM enables a new line of inquiry into how temporal regulation of paralogs influence GRNs in *B. rapa* in which single time point comparisons of differential expression are replaced with temporal pattern analysis to provide a more complete view of the transcriptional network and the pervasiveness of rhythmic gene expression. In particular, this method facilitated the discovery of genes with altered expression patterns independent of expression levels. DiPALM was benchmarked using a publicly available simulated dataset (*Spies et al., 2019*). DiPALM was designed specifically for large time-course datasets in multicellular species that span daylength-scale experiments. This benchmark data consists of simulations that mimic short time-frame cell culture treatments with as little as four timepoints. Despite this, DiPALM performed on par with the top three methods evaluated by Spies et al. (*Supplementary file 8*) and DiPALM remains the only method to use a linear model-based framework that allows for complex contrasts between any number of different treatments. This framework has become the standard for differential expression analysis with packages such as edgeR (*McCarthy et al., 2012*) and DEseq2 (*Love et al., 2014*). In this simulated dataset, a simple pairwise comparison between corresponding timepoints using established differential expression methods was shown to outperform all other methods tested. However, we show that when this 'pairwise'

approach is applied to a real dataset involving mild drought response, it yields no significantly differentially expressed genes, whereas DiPALM detects 3891 genes with differential patterning.

Our data support extensive circadian and diel regulation of the *B. rapa* transcriptome, as documented in Arabidopsis and a few other plant species (*Covington et al., 2008*; *Wilkins et al., 2009*; *Wilkins et al., 2010*; *Li and Zhang, 2015*; *Oakenfull and Davis, 2017*). Our circadian timecourse experiments with 2 hr sampling density provided sufficient resolution to reliably assess rhythmicity for all expressed genes resulting in roughly 74% showing circadian clock regulation in *B. rapa*, consistent with a pervasive role of the circadian clock in regulating diverse aspects of plant physiology. Interestingly, we found that genes retained in multi-copies in *B. rapa* are enriched in network modules that are phased during the day whereas evening and night phased modules are depleted for multi-copy genes. This difference might be related to the dosage sensitivity of processes occurring during the day. Gene ontology enrichment processes for the genes in these daytime phased modules included photosynthesis and abiotic stress response. Given the importance of proper transcriptional regulation of photosynthetic processes and the balance of enzyme components, it is not surprising that there is higher retention of multi-copy genes within these pathways. However, whether these genes are performing similar functions to their orthologous counterparts in Arabidopsis or have acquired new functions that could lead to additional regulation of the photosynthetic process is one exciting avenue of future study. However, it should be noted that in general, evening phased genes are likely to be greatly understudied because most experiments are performed during the day (*Grinevich et al., 2019*), which could artifactually reduce the identification of abiotic stress (and other) genes in our evening modules.

The retention of multi-copy genes that are circadian regulated provided an opportunity to assess the level of retention of transcript abundance patterns among paralogs to look for signs of possible neo- or sub-functionalization. Applying DiPALM to our list of circadian regulated paralogs uncovered evidence for extensive rearrangement of the transcriptional network through the divergence in expression pattern among retained paralogs in *B. rapa* that, in extreme examples, results in paralogs being expressed in antiphase to one another. This demonstrates the genome-wide expansion of phase domains among retained paralogs. The expansion of expression domains is reminiscent of the *PSEUDO-RESPONSE REGULATOR* (*PRR*) and *REVEILLE* (*RVE*) families of circadian clock genes that were retained following WGD as well as tandem duplication events (*Linde et al., 2017*). The *PRR* genes in Arabidopsis have a temporally sequential expression pattern with *PRR9* expressed just after dawn followed by *PRR7*, *PRR5*, *PRR3*, and finally *PRR1/TOC1* in the evening (*Matsushika et al., 2000*). The PRR proteins appear to retain some common functions (e.g. repression of expression of *CCA1*, *LHY*, and *TOC1*) but their diverged expression patterns also allow differential contributions to the circadian network (*Nakamichi, 2011*; *Nakamichi et al., 2012*; *Liu et al., 2013*; *Liu et al., 2016*). This also emphasizes the benefit of applying a network framework to data that associate similarly regulated genes as observed for paralogs with similar expression patterns having their paralogous pairs diverging in expression together.

These changes in phasing among paralogs occur in similar network modules where groups of genes are classified with a particular phase while their paralogous pair-mates exhibit a similar phase difference indicative of common regulatory control. The expansion of expression domains among paralogous pairs provides ample opportunity for neo- and sub-functionalization through new network connections and novel interacting targets now expressed in phase with the pair member with the altered phase of expression. To test this hypothesis and predict the possible network rearrangements that have occurred in *B. rapa* since diverging from its common ancestor with Arabidopsis, we took a GRN approach to model the relationships between TFs and gene expression patterns. The identification of a set of 49 TF *B. rapa* paralogs where one paralog showed significantly more overlap with the Arabidopsis ortholog supports the divergence in network regulation between retained paralogous TFs in *B. rapa*. This is further supported by the overlap of CNSs among target genes in the GRNs. By replacing genes with CNSs in the GRN we were able to identify the more Arabidopsis-like paralog with strong consensus with the original GRN. Among the 35 TFs that overlapped between the CNS and gene expression networks are several genes involved in light signaling (*CRY2*, *PHYA*, *ELF4*, *HYH*, *ZFN1*), photosynthesis (*GLK2*), flowering time (*COL2*, *COL4*, *CDF1*, *CDF2*), abiotic/biotic stress (*RAP2.4*, *STO*, *IBH1*, *TIP*) and the circadian clock (*CDF1*, *CDF2*, *ELF4*, *PRR5*, *PRR9*, *RVE1*, *TIC*, *TOC1*). The divergence in TF among light signaling, photosynthesis, and abiotic stress pathways is consistent with the GO enrichment observed for the daytime phase modules that are

enriched for retained copies. Each of these pathway lists, other than photosynthesis, contains one or more TFs with zinc finger domains. A worthwhile future study would be to investigate whether certain features of TFs make them more likely to neo-functionalize within certain biological pathways. The success of the CNS network approach supports a predictive role of these CNSs for expression dynamics and provides a refined nucleotide sequence space to explore in future studies to associate regulatory elements with specific expression patterns. Associating CNSs with specific gene pattern responses may uncover new regulatory elements or novel combinations of regulatory elements that contribute to the differential regulation of paralogs and their targets, for example in response to drought stress.

In the case of mild drought response, we find that two-copy *B. rapa* paralog pairs with significantly different expression patterns are enriched for pairs in which only one of the copies is drought responsive. This highlights the improved sensitivity gained with time series resolution and the ability to capture critical diel responses to environmental conditions that would otherwise be missed or assumed for all paralogs based solely on homology to Arabidopsis. The significant enrichment of one rather than both members of a paralogous pair being drought responsive also supports neo- or sub-functionalization. This is consistent with observations in Arabidopsis, rice, and poplar, that homeologous genes undergo expression partitioning among tissues (*De Smet and Van de Peer, 2012*; *Langfelder and Horvath, 2012*). This is also seen in the polyploids, *Gossypium hirsitum* and *Tragopogon mirus* (*Adams et al., 2003*; *Buggs et al., 2010*). In Arabidopsis, the majority of duplicated genes show divergent expression (*Blanc and Wolfe, 2004*; *Haberer et al., 2004*), and this is particularly evident among abiotic and biotic stress-responsive genes (*Casneuf et al., 2006*; *Ha et al., 2007*). That significant amino acid sequence variation apparently does not contribute to the divergence between *B. rapa* paralogous TFs reinforces the importance of regulatory element variation in target genes. This raises the question of how two paralogous TFs with the same motif binding affinities can have such diverse targets in the GRN. One possibility is that the presence of new interacting partners at the novel phase of expression could modify binding affinity to either enhance or prevent binding to certain motifs or affect the consequences of TF binding (activation versus repression). Conversely, the lack of critical interacting partners due to a mismatch in phasing could affect target binding and/or regulatory consequence. Further study into the temporal regulation of known binding partners for the divergent TFs is needed.

These findings bring up several questions surrounding the importance of the variation in paralog expression pattern. Are these differential patterns maintained across *B. rapa* morphotypes or is there additional within-species variation? Did these phase differences arise post-genome triplication or were they present in the diploid progenitors that gave rise to *B. rapa*? Different *B. rapa* morphotypes (e.g. leafy vegetable, turnip, and oilseed) exhibit differential circadian clock parameters, as assessed by leaf movement analysis (*Yarkhunova et al., 2016*), strongly supporting additional within-species variation in the transcriptomic network. An examination of the circadian networks across *B. rapa* morphotypes is needed to characterize these differences and begin to associate network plasticity with morphotype-specific traits. Our analysis reveals divergence in drought response among retained paralogs; how do these responses differ in more or less drought-tolerant genotypes? Applying these pattern analysis approaches to pan-transcriptome time-course studies has the potential to identify regulatory elements that contribute to transcriptional network architecture and the evolution of new forms of transcriptional control in polyploids.

## Materials and methods

### Key resources table

| Reagent type (species) or resource | Designation | Source or reference | Identifiers | Additional information |
|---|---|---|---|---|
| Biological sample (*Brassica rapa* subsp. *trilocularis*) | R500 | ABRC | CS28987 | |

*Continued on next page*

*Continued*

| Reagent type (species) or resource | Designation | Source or reference | Identifiers | Additional information |
|---|---|---|---|---|
| Software, algorithm | DiPALM | This paper and CRAN (https://CRAN.R-project.org/package=DiPALM) | | See markdown file on github page (*Greenham, 2020*; https://github.com/GreenhamLab/Brapa_R500_Circadian_Transcriptome) |

## Circadian transcriptome growth conditions

Seeds of *Brassica rapa* subsp. *trilocularis* (Yellow Sarson) R500 were planted in (3.25' x 3.625') pots with a soil mixture of two parts Metro-Mix PX1 + one part Pro-Mix amended with 0.5 mL of Osmocote 18-6-12 fertilizer (Scotts, Marysville, OH). The first photocycle experiment (LD) involved entraining plants under 12 hr light/12 hr dark and constant 20°C (LDHH) for 15 days after sowing (DAS) before transfer to constant light at 20°C (LLHH). The second thermocycle experiment (HC) involved entraining *B. rapa* R500 plants under 24 hr light with 12 hr 20°C and 12 hr 10°C temperature cycles (LLHC) until 15 DAS before transfer to LLHH. Following 24 hr in constant conditions leaf tissue from the youngest leaf was collected and flash-frozen in liquid nitrogen every 2 hr for 48 hr. Lights in the chamber at plant height were ~130 µmol photons $m^{-2}$ $s^{-1}$. Plants were shifted to constant light and temperature (LLHH) for 24 hr before starting the leaf tissue sampling at ZT24. Leaf tissue (~100 mg) from the youngest fully developed leaf was harvested and frozen in liquid nitrogen every 2 hr for 48 hr (ZT24 – ZT72). At each time point, leaf tissue from 10 plants was collected.

## RNA-sequencing library preparations and processing

Leaf tissue was ground to a fine powder using a Retsch Mixer Mill MM 400 (Vendor Scientific, Newtown, PA). The mRNA extraction was performed according to *Greenham et al., 2017* and the strand-specific libraries according to *Wang et al., 2011*. For each leaf sample (~100 mg), 1 mL lysis binding buffer (LBB) was used to resuspend ground tissue. For each of two biological replicates, 200 µL aliquots of LBB lysate from each of five plants were pooled before mRNA isolation. Library size and quality was verified using a 2100-bioanalyzer (Agilent Technologies, Santa Clara, CA). Libraries were indexed and pooled into 12 sample sets and sequenced as 101 bp paired-end reads using Illumina HiSeq2500 (Illumina, San Diego, CA). Raw data have been submitted to GEO (http://ncbi.nlm.nih.gov/geo) under accession number GSE123654 (https://www.ncbi.nlm.nih.gov/geo/query/acc.cgi?acc=GSE123654). The raw fasta reads were filtered using trimmomatic (*Bolger et al., 2014*) with mostly default settings (ILLUMINACLIP:./Tru-Seq3-PE.fa:2:30:10 LEADING:3 TRAILING:3 SLIDING-WINDOW:4:25 MINLEN:50). Reads were aligned to the *B. rapa* R500 genome Brapa_R500_V1.2.fasta (https://genomevolution.org/CoGe/GenomeInfo.pl?gid=52010) using tophat2 (*Kim et al., 2013*; https://ccb.jhu.edu/software/tophat/index.shtml) with the following options: `–library-type` fr-firststrand -I 12000 G R500_v1.6.gff -M `–max-segment-intron` 12000 `–max-coverage-intron` 12000. Sample LD_ZT62_rep2 was identified as an outlier and removed, to avoid overweighting rep1 the rep2 values were imputed by averaging the values of ZT60_rep1 and ZT64_rep1. Raw counts were generated in Subread version 1.6.3 (*Liao et al., 2019*; http://subread.sourceforge.net/) with the following options: -F SAF -M -T 6 –fraction -s 2 p -B -C. Count data was normalized using edgeR (*Robinson et al., 2010*; *McCarthy et al., 2012*; https://bioconductor.org/packages/release/bioc/html/edgeR.html) version 3.22.1 using 'calcNormFactors' and $log_2$ FPKM values were calculated using 'rpkm' with log = TRUE, prior.count = 0.1.

## R package DiPALM

We created an R package that takes a raw count table of RNAseq data and runs differential pattern analysis from time series gene expression data. DiPALM is available through the Comprehensive R Archive Network (CRAN; https://cran.r-project.org/package=DiPALM) or via the Greenham Lab Github page (https://github.com/GreenhamLab/Brapa_R500_Circadian_Transcriptome; *Greenham, 2020*; copy archived at swh:1:rev:9d59bbc84659cbbdffa96413694e01298c9868bb). A sample dataset is provided with the package along with a detailed vignette and manual that describes the analysis pipeline.

## DiPALM benchmarking

A published set of simulated time-course expression datasets (*Spies et al., 2019*) was used for benchmarking. Data consists of eight separate time courses meant to simulate cell culture treatments with samples taken at various times after treatment along with untreated controls at each timepoint. In each dataset, 1200 genes were intentionally perturbed at various intensities using a set of predefined patterns of perturbation. These were taken to be the 'differentially expressed' genes or 'positives' in the test. This benchmarking test was run through DiPALM as well as various other methods tested by Spies et al. Table S1 shows the area under the receiver operating characteristic (AUROC) curve for all tests. The three top-performing methods found by Spies et al. are shown for comparison: splineTC (*Michna et al., 2016*), maSigPro (*Nueda et al., 2014*), and impulseDE2 (*Fischer et al., 2018*). Both the differential pattern (kME) and differential expression (median) detection parts of DiPALM were tested along with a combined score which simply takes the sum of the scores from both parts (combined). The method of running a traditional differential expression analysis on each timepoint separately and then taking the most significant timepoint for each gene was also tested (pairwise).

## Bioinformatic and statistical analysis

The entire analysis pipeline, starting with raw count data, was carried out using the R Statistical Programing Language (*R Development Core Team, 2018*) along with the Rstudio integrated development environment (*R Studio Team, 2015*). A comprehensive R markdown file is available through the Greenham Lab Github page (*Greenham, 2020*; https://github.com/GreenhamLab/Brapa_R500_Circadian_Transcriptome). This analysis script includes all data processing, statistical analysis and plotting that was used for this publication. Additional R packages were used in this analysis, including 'edgeR' (*Robinson et al., 2010*; *McCarthy et al., 2012*), 'stringr' (*Wickham, 2019*), 'ggplot2' (*Wickham, 2016*), 'rain' (*Thaben and Westermark, 2014*), 'WGCNA' (*Langfelder and Horvath, 2008*; *Langfelder and Horvath, 2012*), 'circlize' (*Gu et al., 2014*), and 'pheatmap' (*Kolde, 2019*).

## *B. rapa* paralog expression analysis

For paralog expression comparisons we converted all three-copy paralogs into three two-way comparisons (copy 1 versus copy 2, copy 1 versus copy 3, copy 2 versus copy 3) in order to perform one analysis with the two-copy tests and enable consistency in the interpretation of the results (see line 606 of the markdown file). For the heatmap presented in *Figure 3*, Br1 and Br2 were assigned based on the sign of the summed t-statistics from the limma output (see line 709 of the markdown file).

## *B. rapa* R500 drought RNAseq dataset

For the drought RNAseq analysis we used our previous dataset (*Greenham et al., 2017*) and aligned the data to our new *B. rapa* R500 genome assembly (https://genomevolution.org/CoGe/Organism-View.pl?org_name=Brassica%20rapa) using the same pipeline described for the circadian datasets. These raw counts are available in a file called 'DroughtTimeCourse_CountTable.csv' on the Greenham Lab Github page (*Greenham, 2020*; https://github.com/GreenhamLab/Brapa_R500_Circadian_Transcriptome).

## CNS annotation

Using a list of canonical CNS sequences derived from *Haudry et al., 2013*, the *B. rapa* R500 and TAIR10 genomes were annotated for those CNSs using NCBI BLAST+. A local BLAST database for each reference genome was created using the command:

- makeblastdb -in < reference.fasta> \
- parse_seqids \
- hash_index \
- blastdb_version 5 \
- dbtype 'nucl' \
- title < title >

To get an initial set of CNS alignments, the following command was run for each reference genome:

- blastn -query < cns.fasta> \
- db < reference.fasta> \
- task 'blastn' \
- out < out.csv> \
- outfmt '10 qaccver saccver qlen sstart send sstrand evalue bitscore qcovs' \
- dust 'no' \
- soft_masking 'false' \
- evalue 0.01 \
- num_threads < num_threads>

Filtering was disabled in favor of a different scheme also used in *Yocca et al., 2019*. All alignments with a bitscore of 28.2 were dropped, and alignments with a smaller than 60% coverage of the CNS sequence (BLAST+'s 'qcovs' value) were also dropped. Finally, to ensure the CNS alignments are reasonably unique, all of the alignments for a particular CNS sequence were discarded if they appeared more than once in the TAIR10 reference, or more than three times in the *B. rapa* R500 reference. Since the *B. rapa* genome has undergone a genome triplication event relative to *A. thaliana*, three occurrences were seen as the maximum reasonable amount. Two BED files were generated for each genome containing the coordinates of each resulting alignment.

To associate the resulting CNS alignments with genes in the references, BEDtools was used to find the closest gene to each CNS location:

bedtools closest -s -t all -D a -a < cns.bed> -b < reference.bed>>CNS_prox_genes.txt

cns.bed is one of the two BED files generated in the previous section, and reference.bed is a gene annotation for the respective reference genome. The -s option constraints reported associations to be only on the same strand – that is, CNS alignments and genes must appear on the same strand. -D tells BEDtools to report distances and ensures that the reported distances are signed (negative for occurring before the gene, positive for occurring after, and 0 for being intragenic).

## Motif analysis

Motif analysis was performed using HOMER – specifically, findMotifsGenome.pl. This program requires that two sets of sequences be provided: a set of target sequences to be searched for motifs, and a set of background sequences for comparison to the target sequences to be compared to. Motif analyses were performed on three different target groups.

The first analysis consisted of searching the CNSs against an 'extended' promoter background. For each transcription factor group, the CNSs corresponding to all of the target genes in Ath, Br1, and Br2 separately were pulled and placed into three BED files containing the coordinates of the CNSs in TAIR10 and *B. rapa* R500, respectively. A background set of sequences was then generated for every gene in the TAIR10 and *B. rapa* R500 genomes using BEDtools:

bedtools slop -s -i < reference.bed> -g < reference.genome> -l 2000 r 0 > 2 kb_and_gene.bed

Reference.bed is a gene annotation for the reference genome, and reference.genome is a text file containing the lengths of each chromosome that BEDtools uses to ensure that the coordinates it outputs are valid. This outputs a BED file that annotates a background consisting of a 2 kb promoter region before each gene, as well as the gene itself, to the end of the 3' UTR. This larger background sequence was selected rather than the 2 kb promoter alone since many CNSs occurred in the UTRs and introns, and so using only 2 kb promoters would leave out background sequence relevant to many of the CNSs, potentially skewing the results.

The second analysis searched the 'extended' promoter sequences against themselves. The target set consisted of 'extended' promoters corresponding to target genes in each transcription factor group, and the background consisted of a total list of sequences, including the target group, as per HOMER's recommendations.

The third analysis consisted of a more traditional motif search of promoter regions against promoter-only background. In this case, neither the target nor background sequences contain CDS, introns, or UTRs as with the 'extended' regions defined in the previous two analyses. These promoters were pulled using BEDtools:

bedtools flank -s -i < reference.bed> -g < reference.genome> -l 2000 r 0 > 2 kb_promoters.bed

Everything is identical as above, except that bedtools flank does not include the genes themselves in the output, and only generates locations for promoter regions. As before, promoters corresponding to TF target groups were selected and then analyzed against the entire set of promoters.

HOMER was run using its included plant motif database on all three datasets (*Supplementary files 9–10*). Default parameters were used.

Motif analyses were performed separately for the Ath, Br1, and Br2 target groups. Given that three analyses were performed for each of these target groups, a total of nine motif analyses were performed for each TF group for a total of 108 motif analyses. Only the 'knownMotifs' output of HOMER was considered, which consists of a database search of target and background sequences against known plant motifs with a hypergeometric test to quantify significance. The de novo results were not used. For each of the 108 analyses, the outputs of the 'knownMotifs' analysis were simplified by grouping together found motifs that correspond to the same DNA-binding protein domain. Of each of these domain groups, the best p-value out of all the motifs found for that domain was selected as representative for the entire group. For each TF group, significant (p<0.01) domains were counted for the Ath, Br1, and Br2 target groups, for the CNS, 'extended' promoter, and promoter-only motif analyses.

## Availability of data and materials

The *B. rapa* R500 genome was used for all analyses in this study. It is available through CoGe under the Genome ID: 52010 (https://genomevolution.org/CoGe/GenomeInfo.pl?gid=52010).

Circadian time-course RNA-seq data for *B. rapa* entrained in light cycles (LDHH) and temperature cycle (LLHC) are available through NCBI GEO under accession number GSE123654 (https://www.ncbi.nlm.nih.gov/geo/query/acc.cgi?acc=GSE123654).

Diel time-course RNA-seq under mild drought stress and well-watered conditions is from *Greenham et al., 2017*. Raw data are available through NCBI GEO under accession number GSE90841 (https://www.ncbi.nlm.nih.gov/geo/query/acc.cgi?acc=GSE90841). For this study, the data were re-mapped to the *B. rapa* R500 genome and raw count data are available through a Greenham Lab Github repository (*Greenham, 2020*; https://github.com/GreenhamLab/Brapa_R500_Circadian_Transcriptome). The file is named 'DroughtTimeCourse_CountTable.csv'.

Full details of the data analysis including explanations, code and additional data files can be found in the above Github repository. The analysis is laid out in an R markdown file named 'Brapa_-CircadianTranscriptome_Markdown.Rmd'.

## Acknowledgements

We thank three anonymous reviewers for their helpful suggestions. This work was supported by the National Science Foundation grants IOS-1202779 to KG, IOS-1711662 to RCS, IOS-1547796 to CRM, and by the Rural Development Administration, Republic of Korea Next Generation BioGreen 21, grant number SSAC PJ01327306 to CRM.

## Additional information

### Funding

| Funder | Grant reference number | Author |
| --- | --- | --- |
| National Science Foundation | IOS-1202779 | Kathleen Greenham |
| National Science Foundation | IOS-1711662 | Ryan C Sartor |
| National Science Foundation | IOS-1547796 | C Robertson McClung |
| Rural Development Administration | Next Generation BioGreen 21 grant number SSAC PJ01327306 | C Robertson McClung |

The funders had no role in study design, data collection and interpretation, or the decision to submit the work for publication.

### Author contributions

Kathleen Greenham, Conceptualization, Data curation, Software, Formal analysis, Funding acquisition, Validation, Investigation, Visualization, Methodology, Writing - original draft, Writing - review

and editing; Ryan C Sartor, Conceptualization, Software, Formal analysis, Funding acquisition, Validation, Investigation, Visualization, Methodology, Writing - original draft, Writing - review and editing; Stevan Zorich, Investigation, Methodology; Ping Lou, Conceptualization, Investigation, Methodology, Writing - original draft, Writing - review and editing; Todd C Mockler, Supervision, Methodology, Writing - review and editing; C Robertson McClung, Conceptualization, Supervision, Funding acquisition, Writing - original draft, Project administration, Writing - review and editing

### Author ORCIDs
Kathleen Greenham (iD) http://orcid.org/0000-0001-7681-5263
Ryan C Sartor (iD) http://orcid.org/0000-0002-9621-0824
Stevan Zorich (iD) http://orcid.org/0000-0001-8717-4211
Ping Lou (iD) http://orcid.org/0000-0003-1084-0671
Todd C Mockler (iD) http://orcid.org/0000-0002-0462-5775
C Robertson McClung (iD) https://orcid.org/0000-0002-7875-3614

### Decision letter and Author response
Decision letter https://doi.org/10.7554/eLife.58993.sa1
Author response https://doi.org/10.7554/eLife.58993.sa2

## Additional files

### Supplementary files
• Supplementary file 1. Categorical enrichment of GO Biological Process annotations in WGCNA circadian modules. Coexpression modules were generated for genes in circadian conditions using two different entrainments (LD and HC). Each module was analyzed for categorical enrichment using Biological Process Gene Ontology annotations. Each tab in this file represents a different LD or HC module. The last tab, titled 'Differential_Pattern' represents categorical enrichment results on the list of 1713 genes that show significant pattern changes between LD and HC entrainment conditions.

• Supplementary file 2. DiPALM results for LD versus HC entrainment conditions. Gene expression was evaluated under circadian conditions (constant light and temperature) for 48 hr after plants were entrained in one of the two conditions; LD (12 hr light/12 hr dark) or HC (12 hr 20 °C/ 12 hr 12° C). Using the set of cycling genes that were identified in both datasets, DiPALM was used to detect differential patterning and differential median expression between LD and HC conditions. This dataset reports these results. Column 1 are gene accessions, Column 2 are adjusted p-values for differential patterning and Column 3 are adjusted p-values for differential median expression.

• Supplementary file 3. Categorical enrichment of GO Biological Process annotations in morning-phased and evening-phased multi-copy genes. WGCNA modules generated with cycling genes from a combined LD and HC dataset were analyzed for depletion or enrichment of multi-copy genes. Significant enrichment of multi-copy genes was observed in five morning-phased modules while depletion was observed in four evening-phased modules. This dataset shows the results of Biological Process Gene Ontology categorical enrichment for the set of multi-copy genes in these five enriched modules ('MorningCopied' tab) and the set of multi-copy genes in these four depleted modules ('EveningCopied' tab).

• Supplementary file 4. DiPALM results for paralog comparison in LDHC circadian time course. The LD and HC circadian time-course datasets were combined into one LDHC time course. *B. rapa* paralogous pairs were identified and DiPALM was used to detect differential patterning and differential median expression between paralogs. This dataset reports these results. Column 1 are gene accessions, Column2 are adjusted p-values for differential patterning and Column 3 are adjusted p-values for differential median expression.

• Supplementary file 5. GRN target groups identify which member of *B. rapa* paralog pairs is more Arabidopsis-like. Predicted target groups from GRNs were compared between Arabidopsis (At) and *B. rapa* (Br) networks. Sets of 3 genes containing one At Transcription factor (TF) and its two Br orthologs (denoted B. *rapa* 1 and B. *rapa* 2) were defined. 256 of these TF sets were found between the At and Br GRNs. Tab 'GRN_Comparison_Orthologs' reports results after testing for orthology

between predicted target groups of corresponding At and Br TFs. Column 1 (Gene Accessions) denotes the three accessions for TFs in each set. For simplicity, *B. rapa* 1 was always made to be the Br TF with the most significant overlap in target genes with the At TF. Columns 2, 3, and 4 are the total number of genes in the predicted target groups for At, Br1, and Br2, respectively. Columns 5 and 6 are the number of overlapping genes with the At targets (based on orthology) for Br1 and Br2, respectively. Columns 7 and 8 are the p-values for the significance of overlap with At targets for Br1 and Br2, respectively. Column 9 is the p-value result of a permutation test performed to determine if Br1 is significantly more At-like than Br2. This p-value is meaningless unless Column 7 is also significant. Tab: GRN_Comparison_CNSs reports results after testing for conserved non-coding sequences (CNSs) near genes in predicted target groups of corresponding At and Br TFs. Column 1 (Gene Accessions) denotes the three accessions for TFs in each set. For simplicity, B. *rapa* 1 was always made to be the Br TF with the most significant overlap in target gene CNSs with the At TF. Columns 2, 3, and 4 are the total number of CNSs associated with genes in the predicted target groups for At, Br1, and Br2, respectively. Columns 5 and 6 are the number of overlapping CNSs with the At targets for Br1 and Br2, respectively. Columns 7 and 8 are the p-values for the significance of CNS overlap with At for Br1 and Br2, respectively. Column 9 is the p-value result of a permutation test performed to determine if Br1 is significantly more At-like than Br2. This p-value is meaningless unless Column 7 is also significant. Tab: Significant_Groups summarizes all TF groups where Br1 is significantly more At-like than Br2 based on either the ortholog or CNS comparison. Column 1 lists the three accession numbers for TFs in each group and is ordered based on the target genes orthology comparison (i.e. Br1 is always the most At-like based on orthology of the predicted TF-target genes with predicted At target genes). Column 2 lists the three accessions for TFs in each group and is ordered based on the target genes CNS comparison (i.e. Br1 is always the most At-like based on CNS overlap of the predicted TF-target genes with predicted At target genes). Column 3 lists the p-values for the significance of overlap with At targets and Br1 targets based on orthology. Column 4 indicates if Br1 is significantly more At-like than Br2 based on target gene orthology. Column 5 lists the p-values for the significance of overlap with At targets and Br1 targets based on CNSs. Column 6 indicates if Br1 is significantly more At-like than Br2 based on target gene CNS overlap. Column 7 indicates the 12 TF groups that were used for the motif enrichment analysis (*Figure 4—figure suplement 2*).

• Supplementary file 6. DiPALM results for paralog comparison in diel time course. Gene expression data from control samples in a time course performed in diel conditions (14 hr light 21°C/10 hr dark 18°C) were examined (*Greenham et al., 2017*). *B. rapa* paralogous pairs were identified and DiPALM was used to detect differential patterning and differential median expression between paralogs. This dataset reports these results. Column 1 are gene accessions, Column 2 are adjusted p-values for differential patterning, and Column 3 are adjusted p-values for differential median expression.

• Supplementary file 7. DiPALM results for drought-responsive genes in drought time course. Gene expression data from *Greenham et al., 2017* were examined. Mild drought was imposed and compared to well-watered samples. DiPALM was used to identify differential patterning and differential median expression of genes in response to drought. This dataset reports these results. Column 1 are gene accessions, Column 2 are adjusted p-values for differential patterning, and Column 3 are adjusted p-values for differential median expression.

• Supplementary file 8. DiPALM benchmarking using the area under the receiver operating characteristic (AUROC) curve. DiPALM was evaluated using simulated benchmark datasets (*Spies et al., 2019*). Eight different simulated datasets were generated (rows). Dataset names denote the number of simulated counts, number of simulated replicates, and the number of simulated timepoints. This table also includes the top three performing methods mentioned in Spies et al.: splineTC (*Michna et al., 2016*), maSigPro (*Nueda et al., 2014*), and impulseDE2 (*Fischer et al., 2018*). The method of running a traditional differential expression analysis on each timepoint separately and then taking the most significant timepoint for each gene is also shown (pairwise). Three different DiPALM evaluations were carried out using the pattern-detecting version (kME), the abundance-detecting version (Median) and a combined score using the sum of both.

• Supplementary file 9. CNS annotations for Arabidopsis. Using a list of canonical CNS sequences derived from *Haudry et al., 2013*, the TAIR10 genome was annotated for those CNSs using NCBI

BLAST+. This dataset is in BED file format with six fields as follows: Field 1 denotes the chromosome, Field 2 denotes the start nucleotide position, Field 3 denotes the end nucleotide position, Field 4 denotes the CNS accession (name), Field 5 is a placeholder '1' for a score annotation but is not used in this case, Field 6 denotes the strand.

• Supplementary file 10. CNS Annotations for *Brassica rapa* R500. Using a list of canonical CNS sequences derived from *Haudry et al., 2013*, the *B. rapa* R500 genome was annotated for those CNSs using NCBI BLAST+. This dataset is in BED file format with six fields as follows: Field 1 denotes the chromosome, Field 2 denotes the start nucleotide position, Field 3 denotes the end nucleotide position, Field 4 denotes the CNS accession (name), Field 5 is a placeholder '1' for a score annotation but is not used in this case, Field 6 denotes the strand.

• Transparent reporting form

## Data availability

Sequencing data have been deposited in GEO under accession codes GSE123654.

The following dataset was generated:

| Author(s) | Year | Dataset title | Dataset URL | Database and Identifier |
|---|---|---|---|---|
| Greenham K, Sartor R, Lou P, Mockler TC, McClung CR | 2020 | Expansion of the circadian transcriptome in Brassica rapa supports genome wide diversification of homeolog gene expression patterns | https://www.ncbi.nlm.nih.gov/geo/query/acc.cgi?acc=GSE123654 | NCBI Gene Expression Omnibus, GSE123654 |

The following previously published dataset was used:

| Author(s) | Year | Dataset title | Dataset URL | Database and Identifier |
|---|---|---|---|---|
| Greenham K, Guadagno CR, Gehan MA, Mockler TC, Weinig C, Ewers BE, McClung CR | 2017 | Network analysis identifies temporal regulation of transcriptomic and physiological responses to early drought perception in Brassica rapa. | https://www.ncbi.nlm.nih.gov/geo/query/acc.cgi?acc=GSE90841 | NCBI Gene Expression Omnibus, GSE90841 |

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
