## [Decision Letter]

**Acceptance summary:**

Your careful examination of how the genome-wide triplication in *Brassica rapa* led to a diversification of circadian clock profiles and functions is an important conceptual contribution to the field. The high resolution circadian transcriptome experiments used to characterize the circadian network in *Brassica* and compared to Arabidopsis with a newly developed statistical analysis package will surely become a key tool and resource for others in the field. Of particular note is the investigation of the cycling characteristics of the paralogous genes, with several interesting findings.

**Decision letter after peer review:**

Thank you for submitting your article "Expansion of the circadian transcriptome in *Brassica rapa* and genome-wide diversification of paralog expression patterns" for consideration by *eLife*. Your article has been reviewed by three peer reviewers, and the evaluation has been overseen by Christian Hardtke as the Senior and Reviewing Editor. The reviewers have opted to remain anonymous.

The reviewers have discussed the reviews with one another and the Reviewing Editor has drafted this decision to help you prepare a revised submission.

As the editors have judged that your manuscript is of interest, but as described below that additional analyses are required before it is published, we would like to draw your attention to changes in our revision policy that we have made in response to COVID-19 (https://elifesciences.org/articles/57162). First, because many researchers have temporarily lost access to the labs, we will give authors as much time as they need to submit revised manuscripts. We are also offering, if you choose, to post the manuscript to bioRxiv (if it is not already there) along with this decision letter and a formal designation that the manuscript is "in revision at *eLife*". Please let us know if you would like to pursue this option. (If your work is more suitable for medRxiv, you will need to post the preprint yourself, as the mechanisms for us to do so are still in development.)

As you can see from the individual reviews below, all reviewers are in principle positive about your paper. However, they all have a number of comments/suggestions for improvement, which I would like to ask you to address to the best of your capacity.

Reviewer #1:

In this interesting work, Greenham et al., examine how the genome wide triplication in *Brassica rapa* led to a diversification of circadian clock profiles and functions. The work is carefully done, with two high resolution circadian transcriptome experiments used to characterize the circadian network in *Brassica* and compared to Arabidopsis. The authors develop a novel statistical analysis package, Differential Pattern Analysis via Linear Models (DiPALM), that assigns statistical significance to genes based on pattern changes between two or more experimental condition, which could also be of general interest. Of particular note is the investigation of the cycling characteristics of the paralogous genes, with several interesting findings. I have the following comments:

I was a bit confused by the analysis of the two datasets (LD and HC entraining cycles). These datasets are considered so similar they can be pooled in later analysis, but the LD data has 14 modules whilst the HC has 10. The DiPalm analysis reveals that 11% of cycling genes have entrainment dependent cycling. The authors state “The 11% of cycling genes with entrainment-dependent cycling patterns are very interesting but further analysis in this area is not within the intended scope of this manuscript”. For me to be comfortable with the combining of these datasets it would be great to know a little more about these differences (or if this is going to be analysed in a different paper, a discussion of how these 11% of genes were treated when the two datasets were pooled would be useful. Currently it is written “Therefore, for all subsequent references to the RNA-seq dataset we combined LD and HC entrainments and included the LD/HC factor as a covariate in the linear model.” But I wasn't sure what the LD/HC factor was?)

Additionally, the RNA-seq data suggested 76 percent of genes are circadian regulated. This appears much higher than the proportion normally discussed in Arabidopsis (20-30%). Could the authors examine what gene categories are cycling in *Brassica* that are not cycling in Arabidopsis? It would be interesting to find out more about the small fraction of genes that are not cycling. Finally, is there any possible independent validation or comparison to previous methods of characterising the differences between oscillating systems that could be carried out for DiPalm? Currently it is hard to know how robust the results of the DiPalm analysis are.

Reviewer #2:

In this work, Greenham et al. provides evidence of genetic diversification of transcription factors following duplication events. The study uses transcriptomics of Arabidopsis and *Brassica rapa* measured during diel and circadian rhythms to determine potential neo-functionalization of paralogous genes.

Although intellectually arduous for me (I'm not a specialist of evolution) the paper is relatively easy to read and very well written. The techniques (including new ones (DiPALM) developed by the group) used to derive hypothesis are convincing. The new transcriptomes represent important datasets being available to the plant biology community.

Even though I think the terminology can be misleading, an important outcome to my eyes is the classification (and the methodology to retrieve them) of *B. rapa* genes into "more Arabidopsis-like" and "Less Arabidopsis-like genes". This classification is then used to show that a significantly greater proportion of paralogs which circadian pattern is different tend to be modified by this abiotic factor. This opens interesting investigation path to show that different environmental factors may have different evolutionary shaping effect of GRNs.

I have very few critiques being more remarks/thoughts.

First, I slightly regret that the thinking goes from Arabidopsis to *B. rapa*. Can the Authors try to get back to At? For instance, I wonder if it might be possible to speculate concerning the Arabidopsis TF features (gene family, protein sequence, protein partners, target CRE composition/sequence… anything) that may "push" towards a gene neo-functionalization?

Is there any other transcriptomic dataset available to test another biotic or abiotic factor and see if what has been found for drought is general of peculiar?

Reviewer #3:

The authors report results from an integrated study of two comprehensive datasets that are designed to test effects of photocycle and thermocycle on gene expression changes in *Brassica rapa*. They found a substantial fraction (75%) of expressed genes was regulated by one or both conditions, and the majority of them overlapped between the two. They further analyzed these transcriptional modules and identified subsets of transcriptional factors, including key circadian clock regulators, regulate expression networks or domains through conserved noncoding sequences (CNS) of the target or co-regulated genes. *B. rapa* underwent three rounds of genome duplication, relative to *A. thaliana*. They further tested if duplicate genes show higher levels of expression divergence under these conditions. While the majority are co-regulated, some paralogs showed diverged expression phases, including those in stress responses such as drought. These resources are valuable for the genomics and circadian biology communities. The overall data interpretation is reasonable, although some areas can be tightened up to make it concise. The manuscript can be improved by addressing the following comments.

1) The Abstract does not seem to capture the major findings about the conservation of circadian transcriptional networks, instead, places an emphasis on divergence of expression between paralogs. This over-emphasized approach has made some data rearranged and confused in the Abstract. For example, the network regulation by transcription factors via CNS is part of conservation of the system. It was placed after diverged stress response genes, as if these paralogs with diverged expression were regulated by CNS, which is misleading. CNS would infer co-regulation. In another place, they used diurnal and diel in the same sentence, which has confounded the impact of their beautiful data with published "diel" data. Apparently, the latter was used only for one part of comparative analysis.

2) Results section paragraph two and Figure 1. LD genes are grouped into 14 modules, and HC genes are in 10 modules. Perhaps additional details should be provided. For example, overlapping modules are largely in phase or out of phase; expression levels are similar or different; why are there 14 modules in one and 10 in another condition? The discordant data (11%, paragraph four) are excluded for further analysis. Could these discordant data be relevant to expression divergence of paralogs or simply noise?

3) The term "eigengene" is introduced without a clear definition. In some cases, its use is at least confusing, singular vs. plural, etc. The concept should be introduced in the beginning and used precisely in the context of results. In addition, it is unclear how effective the module membership (kME) is implicated for network analysis?

4) Figure 2 and paragraph seven, the division of two expression groups (high and low) was not clearly defined. Different retention rates of duplicate genes in morning-phased and evening-phased genes are interesting and should be discussed in the paper. In addition, why should the response to abiotic stimulus genes be enriched only in the morning phased genes? How about the biotic response genes? "…consistent with a phase at dusk," how so? The images and labels in most figure panels are too small to be legible.

5) Results paragraph eight and Figure 3. The definition of "exDif" and "pDif" is very confusing and should be clarified, which could also contribute to a confusion in Figure 3 (see below).

“The expansion of expression domains is reminiscent of the PSEUDO-RESPONSE REGULATOR (PRR) and REVEILLE (RVE) families of circadian clock genes that were retained following WGD as well as tandem duplication events (Linde et al., 2017). […]The PRR proteins appear to retain some common functions but their diverged expression patterns results in differential contributions to the circadian network (Nakamichi, 2011; Nakamichi et al., 2012; Liu et al., 2013; Liu et al., 2016).”. The part is heavily cited with published data and belongs to the Discussion.

6) Figure 3. The data need clarification, especially with the statement in the figure legend, "Three-copy paralogs were split into three 2-way comparisons". How are three paralogs split into two, Br1 and Br2? Also, this sentence does not make sense, "Each line of the heat map for each block corresponds to a paralogous pair." Each line (row?) = a pair? This part of data should be revisited and interpreted with caution.

7) Data in Figure 4 are interesting and described more clearly in the text than in the Figure, partly because figure panels were not sequentially labeled, neither were the legends. This should be fixed. "For all 12 TFs tested, we found 3-5 fold greater overrepresentation of motifs in CNS…" There are two obvious questions, what are these CNS, and what are the motifs in CNS? I guess they depend on TFs. Perhaps they can give some examples of CNS with respect to the TFs as described in Figure 4C?

8) Expression pattern (but not expression level) divergence of paralogs in stress (drought) response is interesting. Perhaps they should cite and discuss some old references (Casneuf et al., 2006; Ha et al. 2007) relevant to preferential retention of stress responsive genes and diurnal regulation of stress responsive genes in different ecotypes (Miller et al. 2015, Nature Communications)? This may help interpret biological relevance to the conservation and divergence of paralogs in paleopolyploids.

9) A general problem for data description is that the authors appear to have done statistical tests for most analyses, but it is unclear what the tests and p values are, or they are not cited in the relevant text. In addition, although the authors emphasized triplication of *B. rapa* genome, only two paralogs were used in most, if not all, analyses. Why?

10) Overall presentation and Discussion. The paper seems to lack some important take-home messages about the answers to a key question as the authors wanted to address, "Have these retained paralogs diversified in function and contributed to robustness and flexibility in the circadian clock?" For example, most paralogs are co-regulated by the photocyle and thermocycle conditions, while some such as stress responsive genes are diverged. What are the possible reasons and how will they impact plant growth and development during evolution and domestication? The current Discussion appears to focus on technical advance (e.g., DiPALM) and reiterate some results with a little effort on providing biological perspectives and insights.

The paper is relatively long and can be improved to reduce redundant and background descriptions (throughout the results) with focus and clarity. In the Results, some descriptions and narratives can be condensed or moved into Materials and methods or Discussion.

---

## [Author Response]

Reviewer #1:In this interesting work, Greenham et al., examine how the genome wide triplication in Brassica rapa led to a diversification of circadian clock profiles and functions. The work is carefully done, with two high resolution circadian transcriptome experiments used to characterize the circadian network in Brassica and compared to Arabidopsis. The authors develop a novel statistical analysis package, Differential Pattern Analysis via Linear Models (DiPALM), that assigns statistical significance to genes based on pattern changes between two or more experimental condition, which could also be of general interest. Of particular note is the investigation of the cycling characteristics of the paralogous genes, with several interesting findings. I have the following comments:I was a bit confused by the analysis of the two datasets (LD and HC entraining cycles). These datasets are considered so similar they can be pooled in later analysis, but the LD data has 14 modules whilst the HC has 10. The DiPalm analysis reveals that 11% of cycling genes have entrainment dependent cycling. The authors state “The 11% of cycling genes with entrainment-dependent cycling patterns are very interesting but further analysis in this area is not within the intended scope of this manuscript”. For me to be comfortable with the combining of these datasets it would be great to know a little more about these differences (or if this is going to be analysed in a different paper, a discussion of how these 11% of genes were treated when the two datasets were pooled would be useful. Currently it is written “Therefore, for all subsequent references to the RNA-seq dataset we combined LD and HC entrainments and included the LD/HC factor as a covariate in the linear model.” But I wasn't sure what the LD/HC factor was?)

We have expanded the explanation of how these two datasets were handled in the analysis given the 11% of genes that showed differences between the two entrainment conditions. The main text now reads:

“For the remainder of analyses with these datasets, we have combined LD and HD to increase our statistical power by having four replicates per time point rather than two. One significant advantage of a linear model-based framework is the ability to account for any identified effect. We make use of this feature by modeling any effects between LD and HC as a covariate. In other words, DiPALM gives us the ability to combine these datasets while still accounting for any differences between LD and HC. This combined dataset provides more statistical power for further analysis, particularly for the 89% of genes that show no significant difference between LD and HC. For the other 11% that do have differences, these differences are taken into account and will not adversely affect the results”.

Additionally, the RNA-seq data suggested 76 percent of genes are circadian regulated. This appears much higher than the proportion normally discussed in Arabidopsis (20-30%). Could the authors examine what gene categories are cycling in Brassica that are not cycling in Arabidopsis? It would be interesting to find out more about the small fraction of genes that are not cycling. Finally, is there any possible independent validation or comparison to previous methods of characterising the differences between oscillating systems that could be carried out for DiPalm? Currently it is hard to know how robust the results of the DiPalm analysis are.

We completely agree that this difference with previous studies in Arabidopsis is quite striking; however, there are some technical reasons that may account for some of the differences and make it difficult to draw too many conclusions. The numbers in Arabidopsis are based on microarray experiments that introduce additional variation due to the nature of probe-based assays that can affect the robustness of cycling gene transcript profiles and ultimately the assignment of cycling or not-cycling. Another reason for our higher cycling call is likely that our higher sampling resolution with 2h rather than the 4h that was used in the Arabidopsis studies affords greater statistical power. Lastly, various programs use slightly different methods for calling transcript abundance patterns as rhythmic and the choice of p-value greatly influences how many genes are called cycling or not. Due to all these variables, we chose to refrain from characterizing genes that are “not-cycling” because we are not completely confident in our ability to classify these. Using a conservative p-value cutoff we can be fairly confident in our call of cycling but inevitably we will miss some that we do not want to characterize as not-cycling.

In terms of DiPALM, the reviewer makes an excellent point. We were able to find a recent study testing various packages designed to capture gene expression responses to short time-course cell-culture treatments. While the simulated data in this study are not exactly what DiPALM was designed to detect, it did allow us to test the robustness against other methods on as little as 4 timepoints and we were pleased to see it performed on par with the top 3 methods tested. We have included the results from this in Supplementary file 8 and have included the following text in the manuscript:

The Discussion section now reads:

“DiPALM was benchmarked using a publicly available simulated dataset (Spies et al., 2017). DiPALM was designed specifically for large time-course datasets in multicellular species that span daylength-scale experiments. This benchmark data consists of simulations that mimic short time-frame cell-culture treatments with as little as 4 timepoints. Despite this, DiPALM performed on par with the top 3 methods evaluated by Spies et al. (Supplementary file 8) and DiPALM remains the only method to use a linear model-based framework that allows for complex contrast between any number of different treatments. This framework has become the standard for differential expression analysis with packages such as edgeR (McCarthy et al., 2012) and DEseq2 (Love et al., 2012). In this simulated data a simple pairwise comparison between corresponding timepoints using established differential expression methods was shown to outperform all other methods tested. However, we show that when this “pairwise” approach is applied to a real dataset involving mild drought response, it yields no significantly differentially expressed genes, while DiPALM detects 3891 genes with differential patterning.“

Reviewer #2:In this work, Greenham et al. provides evidence of genetic diversification of transcription factors following duplication events. The study uses transcriptomics of Arabidopsis and Brassica rapa measured during diel and circadian rhythms to determine potential neo-functionalization of paralogous genes.Although intellectually arduous for me (I'm not a specialist of evolution) the paper is relatively easy to read and very well written. The techniques (including new ones (DiPALM) developed by the group) used to derive hypothesis are convincing. The new transcriptomes represent important datasets being available to the plant biology community.Even though I think the terminology can be misleading, an important outcome to my eyes is the classification (and the methodology to retrieve them) of B. rapa genes into "more Arabidopsis-like" and "Less Arabidopsis-like genes". This classification is then used to show that a significantly greater proportion of paralogs which circadian pattern is different tend to be modified by this abiotic factor. This opens interesting investigation path to show that different environmental factors may have different evolutionary shaping effect of GRNs.I have very few critiques being more remarks/thoughts.First, I slightly regret that the thinking goes from Arabidopsis to B. rapa. Can the Authors try to get back to At? For instance, I wonder if it might be possible to speculate concerning the Arabidopsis TF features (gene family, protein sequence, protein partners, target CRE composition/sequence… anything) that may "push" towards a gene neo-functionalization?

We have expanded the text to describe the pathways that the 35 overlapping set of TFs are involved in as well as an interesting finding that most of these pathways contain one or more TF in the zinc finger family. It is interesting to speculate whether specific TF features within a pathway (ie. Light signaling) makes it more likely to neo-functionalize. Thank you for the great suggestion and question.

Text added to the Discussion:

“Among the 35 TFs that overlapped between the CNS and gene expression networks are several genes involved in light signaling (*CRY2, PHYA, ELF4, HYH, ZFN1*), photosynthesis (*GLK2,*), flowering time (*COL2, COL4, CDF1, CDF2*), abiotic/biotic stress (*RAP2.4, STO, IBH, TIP*) and the circadian clock (*CDF1, CDF2, ELF4, PRR5, PRR9, RVE1, TIC, TOC1*). The divergence in TF among light signaling, photosynthesis and abiotic stress pathways is consistent with the GO enrichment observed for the day time phase modules that are enriched for retained copies (Figure 2D). Each of these pathway lists, other than photosynthesis, contain one or more TFs with zinc finger domains. A worthwhile future study would be to investigate whether certain features of TFs make them more likely to neo-functionalize within certain biological pathways.”

Is there any other transcriptomic dataset available to test another biotic or abiotic factor and see if what has been found for drought is general of peculiar?

Unfortunately, we are not aware of another biotic or abiotic time course expression study in *B. rapa* to which we could compare our drought findings. We wish there was! We are generating a time course experiment following cold stress that hopefully will allow us to answer this question in the future.

Reviewer #3:The authors report results from an integrated study of two comprehensive datasets that are designed to test effects of photocycle and thermocycle on gene expression changes in Brassica rapa. They found a substantial fraction (75%) of expressed genes was regulated by one or both conditions, and the majority of them overlapped between the two. They further analyzed these transcriptional modules and identified subsets of transcriptional factors, including key circadian clock regulators, regulate expression networks or domains through conserved noncoding sequences (CNS) of the target or co-regulated genes. B. rapa underwent three rounds of genome duplication, relative to *A. thaliana*. They further tested if duplicate genes show higher levels of expression divergence under these conditions. While the majority are co-regulated, some paralogs showed diverged expression phases, including those in stress responses such as drought. These resources are valuable for the genomics and circadian biology communities. The overall data interpretation is reasonable, although some areas can be tightened up to make it concise. The manuscript can be improved by addressing the following comments.1) The Abstract does not seem to capture the major findings about the conservation of circadian transcriptional networks, instead, places an emphasis on divergence of expression between paralogs. This over-emphasized approach has made some data rearranged and confused in the Abstract. For example, the network regulation by transcription factors via CNS is part of conservation of the system. It was placed after diverged stress response genes, as if these paralogs with diverged expression were regulated by CNS, which is misleading. CNS would infer co-regulation. In another place, they used diurnal and diel in the same sentence, which has confounded the impact of their beautiful data with published "diel" data. Apparently, the latter was used only for one part of comparative analysis.

The Abstract has been rewritten with the reviewer’s comments in mind. In terms of the CNS analysis, we are suggesting that the divergence in paralog expression may be in part due to differences in CNS retention. In some cases the *B. rapa* paralogs both retain the CNS that are found in their Arabidopsis orthologous counterpart but in many cases they are not. Often we see paralogs with unique CNS, some or all of which may be found in the Arabidopsis ortholog.

2) Results section paragraph two and Figure 1. LD genes are grouped into 14 modules, and HC genes are in 10 modules. Perhaps additional details should be provided. For example, overlapping modules are largely in phase or out of phase; expression levels are similar or different; why are there 14 modules in one and 10 in another condition? The discordant data (11%, paragraph four) are excluded for further analysis. Could these discordant data be relevant to expression divergence of paralogs or simply noise?

See above response to reviewer 1 on treatment of 11%.

A rough summary of phase relationships between modules is shown in the Figure 1 Circos plot. The text and legend now read:

“The width of the ribbon signifies the number of genes in common between the connected modules and the color represents the Pearson correlation coefficient between the eigengenes of the two datasets.”

3) The term "eigengene" is introduced without a clear definition. In some cases, its use is at least confusing, singular vs. plural, etc. The concept should be introduced in the beginning and used precisely in the context of results. In addition, it is unclear how effective the module membership (kME) is implicated for network analysis?

We have provided a definition in the text: (first principle component of the expression matrix for each module). The kME is used as a representation of the expression pattern of a gene. Rather than using a single expression value as input into a linear model we use the kME that captures the full set of time course expression values. We have explained the use of kME in the Introduction. Its use can also be seen in the DiPALM documentation found with the package and on the GitHub page (https://github.com/GreenhamLab/Brapa_R500_Circadian_Transcriptome).

4) Figure 2 and paragraph seven, the division of two expression groups (high and low) was not clearly defined. Different retention rates of duplicate genes in morning-phased and evening-phased genes are interesting and should be discussed in the paper. In addition, why should the response to abiotic stimulus genes be enriched only in the morning phased genes? How about the biotic response genes? "…consistent with a phase at dusk," how so? The images and labels in most figure panels are too small to be legible.

We have clarified how the high and low groups were generated:

“To test whether this is the case, we separated all the cycling 2- and 3-copy paralogs into the highest and lowest expressed copies based on the average expression level across all time points. In the case of 3-copy paralogs, we only included the highest and lowest expressed genes in the analysis.”

We also included a possible explanation for why response to abiotic stimulus might only be enriched in the morning phased genes:

“However, it should be noted that in general, evening phased genes are likely to be greatly understudied because most experiments are performed during the day (Grinevich et al., 2019), which could artefactually reduce the identification of abiotic stress (and other) genes in our evening modules.”

5) Results paragraph eight and Figure 3. The definition of "exDif" and "pDif" is very confusing and should be clarified, which could also contribute to a confusion in Figure 3 (see below).

We’ve removed the use of the exDif and pDif acronyms from the text and replaced them with more accurate descriptions to avoid unnecessary confusion. We have also clarified the legend for Figure 3 to include the following additional text:

“Heat map of the results from the DiPALM differential pattern analysis (pDif) clustering showing the changes in expression pattern for paralogous pairs. The Br1 and Br2 heat map blocks for each module contain matching paralogous pairs; for example, the gene in the first line of Br1 and Br2 heatmaps for module pDif_01 are paralogs. The blocks have been stacked on top of each other to be able to compare the change in pattern across the time course.”

“The expansion of expression domains is reminiscent of the PSEUDO-RESPONSE REGULATOR (PRR) and REVEILLE (RVE) families of circadian clock genes that were retained following WGD as well as tandem duplication events (Linde et al., 2017). […]The PRR proteins appear to retain some common functions but their diverged expression patterns results in differential contributions to the circadian network (Nakamichi, 2011; Nakamichi et al., 2012; Liu et al., 2013; Liu et al., 2016).”. The part is heavily cited with published data and belongs to the Discussion.

We have moved this section to the Discussion.

6) Figure 3. The data need clarification, especially with the statement in the figure legend, "Three-copy paralogs were split into three 2-way comparisons". How are three paralogs split into two, Br1 and Br2? Also, this sentence does not make sense, "Each line of the heat map for each block corresponds to a paralogous pair." Each line (row?) = a pair? This part of data should be revisited and interpreted with caution.

We have clarified the legend for Figure 3 and included a new section in the Materials and methods (*B. rapa* paralog expression analysis) that explains the assignment of Br1 and Br2.

7) Data in Figure 4 are interesting and described more clearly in the text than in the Figure, partly because figure panels were not sequentially labeled, neither were the legends. This should be fixed. "For all 12 TFs tested, we found 3-5 fold greater overrepresentation of motifs in CNS…" There are two obvious questions, what are these CNS, and what are the motifs in CNS? I guess they depend on TFs. Perhaps they can give some examples of CNS with respect to the TFs as described in Figure 4C?

We have clarified the legend for Figure 4 and the panels are now described sequentially in the text. For the CNS motif analysis; in Arabidopsis, every gene has a unique set of CNS. They do not repeat throughout the genome. For this reason, we ran the motif analysis to try to associate similar sequences within the CNS in the target groups. Because GENIE3 does not assign a direction of regulation to target groups (positive and negative regulation are treated equally) the expression patterns of target groups for a single TF can be quite diverse and as a result there is a lot of overlap of motifs in each target group. For the purposes of this study we chose to simply emphasize the enrichment of motifs within CNS compared to promoter regions. It will be very interesting to classify the gene expression patterns within each target group and identify specific motifs that are associated within those patterns. Given the current length of the manuscript we thought it best to leave this for a future study.

8) Expression pattern (but not expression level) divergence of paralogs in stress (drought) response is interesting. Perhaps they should cite and discuss some old references (Casneuf et al., 2006; Ha et al., 2007) relevant to preferential retention of stress responsive genes and diurnal regulation of stress responsive genes in different ecotypes (Miller et al., 2015, Nature Communications)? This may help interpret biological relevance to the conservation and divergence of paralogs in paleopolyploids.

Thank you for the suggestion. We have updated the text:

“The significant enrichment of one rather than both members of a paralogous pair being drought responsive also supports neo- or sub-functionalization (Figure 5). This is consistent with observations in Arabidopsis, rice, and poplar, that homeologous genes undergo expression partitioning among tissues (De Smet and Van de Peer, 2012). This is also seen in the polyploids, *Gossypium hirsitum* and *Tragopogon mirus* (Adams et al., 2003; Buggs et al., 2010). In Arabidopsis, the majority of duplicated genes show divergent expression (Blanc and Wolfe, 2004; Haberer et al., 2004), and this is particularly evident among abiotic and biotic stress-responsive genes (Casneuf et al., 2006; Ha et al., 2007).”

9) A general problem for data description is that the authors appear to have done statistical tests for most analyses, but it is unclear what the tests and p values are, or they are not cited in the relevant text. In addition, although the authors emphasized triplication of B. rapa genome, only two paralogs were used in most, if not all, analyses. Why?

We have gone through and included p-value cutoffs throughout the text where they had been missing. We did use the 3-copy paralogs in our analysis by converting them into three 2-way comparisons to avoid having to develop a new statistical tool to compare expression dynamics between the 3 copies. Additionally, applying a different method on 2-copy and 3-copy paralogs would make the interpretation of the analysis challenging. By reducing the 3-copies into 2-way comparisons we could apply the same analysis to all paralogs. To clarify this in the text we have added a section to the Materials and methods to explain the paralog analysis.

10) Overall presentation and Discussion. The paper seems to lack some important take-home messages about the answers to a key question as the authors wanted to address, "Have these retained paralogs diversified in function and contributed to robustness and flexibility in the circadian clock?" For example, most paralogs are co-regulated by the photocyle and thermocycle conditions, while some such as stress responsive genes are diverged. What are the possible reasons and how will they impact plant growth and development during evolution and domestication? The current Discussion appears to focus on technical advance (e.g., DiPALM) and reiterate some results with a little effort on providing biological perspectives and insights.

We have edited the Discussion with these comments in mind.

The paper is relatively long and can be improved to reduce redundant and background descriptions (throughout the results) with focus and clarity. In the Results, some descriptions and narratives can be condensed or moved into Materials and methods or Discussion.

We have shortened the Results as best we could while maintaining clarity and moved redundant sections to the Discussion.